# Low-k nano-dielectrics facilitate electric-field induced phase transition in high-k ferro-electric polymers for sustainable electro-caloric refrigeration

Qiang Li[1], Luqi Wei[2], Ni Zhong [2], Xiaoming Shi[3], Donglin Han [1], Shanyu Zheng[1], Feihong Du[1], Junye Shi[1], Jiangping Chen[1], Houbing Huang [3], Chungang Duan[2] & Xiaoshi Qian [1,4] ✉

Ferroelectric polymer-based electrocaloric effect may lead to sustainable heat pumps and refrigeration owing to the large electrocaloric-induced entropy changes, flexible, lightweight and zero-global warming potential. Herein, low-k nanodiamonds are served as extrinsic dielectric fillers to fabricate polymeric nanocomposites for electrocaloric refrigeration. As low-k nanofillers are naturally polar-inactive, hence they have been widely applied for consolidate electrical stability in dielectrics. Interestingly, we observe that the nanodiamonds markedly enhances the electrocaloric effect in relaxor ferroelectrics. Compared with their high-k counterparts that have been extensively studied in the field of electrocaloric nanocomposites, the nanodiamonds introduces the highest volumetric electrocaloric enhancement (~23%/vol%). The resulting polymeric nanocomposite exhibits concurrently improved electrocaloric effect (160%), thermal conductivity (175%) and electrical stability (125%), which allow a fluid-solid coupling-based electrocaloric refrigerator to exhibit an improved coefficient of performance from 0.8 to 5.3 (660%) while maintaining high cooling power (over 240 W) at a temperature span of 10 K.

Electricity consumption for cooling and heating accounts for more than 30% of total household electricity consumption worldwide[1]. These thermal management systems use refrigerants that exhibit high global warming potential (GWP) and their leakage during operation cannot be prevented[2–4]. The most common refrigerants used in these systems are hydrofluorocarbons (HFCs), which have GWPs that can be thousands of times higher than carbon dioxide[5,6]. According to the International Energy Agency (IEA), in 2019, refrigeration and air conditioning systems accounted for around 10% of global greenhouse gas (GHG) emissions[7]. In particular, the residential and commercial building sector was responsible for approximately 60% of these emissions[8]. To address this issue, many countries have started to phase out HFCs (and other high-GWP refrigerants) and develop lower-GWP alternatives[9–12]. In addition, the development of compressor-free technologies has become an important issue in the current development of new refrigeration and heat pump technologies[13–18]. Among

[1]State Key Laboratory of Mechanical System and Vibration, Interdisciplinary Research Center, Institute of Refrigeration and Cryogenics, and MOE Key Laboratory for Power Machinery and Engineering, School of Mechanical Engineering, Shanghai Jiao Tong University, Shanghai 200240, China. [2]Key Laboratory of Polar Materials and Devices, Ministry of Education, Shanghai Center of Brain-inspired Intelligent Materials and Devices, East China Normal University, Shanghai 200241, China. [3]School of Materials Science and Engineering and Advanced Research Institute of Multidisciplinary Science, Beijing Institute of Technology, 100081 Beijing, China. [4]Shanghai Jiao Tong University ZhongGuanCun Research Institute, Liyang 213300, China. ✉e-mail: xsqian@sjtu.edu.cn

them, new solid-state technologies such as electrocaloric (EC) refrigeration (based on the electrocaloric effect, ECE) are known for their potential capability to approach a low-carbon, highly efficient future of refrigeration[11,19,20].

EC refrigeration utilized the charging and discharging processes of dielectric capacitors to achieve the refrigeration cycle with high theoretical efficiency. After the invention of EC ceramics[21] and polymers[22] in 2006-2008, it quickly attracted widespread attention around the world[23–25]. Existing studies on EC devices are featuring high energy reversibility (>90%)[20], high energy recovery efficiency of the driving electric field (>80%)[26], simple driving method[27], compact device configurations[28], and high theoretical specific cooling power density[25,26,29,30]. In addition, the widely-used EC ceramics and polymers could fit into the existing processes for mass production, i.e., multilayer ceramic capacitor (MLCC)[31,32] and roll-to-roll (R2R)[33,34], therefore the EC refrigeration has been considered as one of the promising solid-state refrigeration technologies[27,35–38]. Currently, both ceramic and polymer-based EC devices have been demonstrated by designing the active electrocaloric regeneration (AER) cycles, which were achieved by fluid-solid and soli-solid conjugated heat transfer[39–41]. These designs could reach large temperature spans that have the potential to be comparable to that of conventional vapor compress refrigeration. The other major breakthrough of the EC refrigerators was the demonstration of the electrostatically driven polymer-based, thin-film refrigerators that could be applied in many niche applications where space and weight-limit are tight and the conventional technologies cannot fit in[27,41]. Utilizing only a limited amount of the active cooling material, the cascaded thin-film EC device[41] achieved an impressive cooling power of 78.5 mW·cm$^{-2}$. When a large quantity of EC materials is required for further improved total cooling power, the EC working body would be inevitably thicker and the time for the complete heat transfer would be much longer. Therefore, the research field asks for the EC materials to possess larger ECEs and thermal conductivities than that of the widely adopted EC materials, in order to maintain or even increase the cycling frequency for an enhanced cooling power[42].

Although the EC polymers enabled many unique cooling applications owing to their inherited flexibility, low density, and electrical stability, they are also known for their low thermal conductivity (0.2 W·m$^{-1}$·K$^{-1}$) that poses a challenge to the actual refrigeration devices to achieve its theoretical efficiency[43,44]. Moreover, the existing EC polymer still requires an extremely high electric field to achieve satisfactory refrigeration performance[11]. Polymer-rich nanocomposites represent a feasible approach for enhancing both the ECE and thermal conductivity while maintaining the other features as soft materials. Currently, the majority of the fillers employed are nanoparticles of ferroelectric inorganic oxides with high permittivity (k), resulting in an amplified distortion of the electric field at the polymer-oxide interface, and promoting the interfacial polarization[45–48]. This phenomenon was believed to consequently reinforce the ECE[47,48]. Chen et al. [23] improved the polarization and ECE by means of preparing (P(VDF−TrFE−CFE))/ZrO$_2$ nanocomposites. The permittivity of ZrO$_2$ nanoparticles ($\varepsilon$ ~ 30) is low compared to many oxides, but is comparable to that of base terpolymer ($\varepsilon$ ~ 45). The small difference between the permittivity is insignificant to induce large internal field-distortion at the organic-inorganic interface.

In this work, we adopted unconventional, low-k nanodielectrics (nanodiamond, ND) in the commercially available EC polymer poly(vinylidene fluoride-trifluoroethylene-chlorofluoroethylene) (P(VDF-TrFE-CFE), 65/35/7.5 mol%), with a hypothesis to enhance the ECE, electrical stability, and thermal conductivity concurrently in the nanocomposites. The low-k nanoparticles were extensively incorporated in the non-polar polymeric dielectrics to reduce their dielectric response[49]. With a low content of ND in the relaxor ferroelectric polymer, we observed a starkly distinguished phenomenon, in which the ND-incorporated nanocomposite exhibited a strongly enhanced polarization and ECE. These enhancements are even more effective compared to their high-k counterparts, i.e., the inclusion of 2.6 vol% of ND (resulting in a nanocomposite referred to as T-ND-2.6%) could increase the ECE for 60% at an electric field of 100 MV·m$^{-1}$, which is corresponding to a 23%/vol%-filler. The value (volumetric enhancement of ECE) is nearly 70% higher than the best-reported high-k fillers[47]. Considering the low content of the ND, the thermal conductivity of the nanocomposite increased by 75%. With the in-situ XRD, we demonstrated that the enhancement of the ECE is due to the reduction of the energy barrier of the nonpolar-polar phase transition in the nanocomposites. From the Electrostatic Force Microscopy (EFM) test, we directly observed a significant build-up of potential at the ND-polymer interface, which may explain the improved polarization of the nanocomposites. Owing to the concurrently enhanced ECE and thermal conductivity, we developed a model of EC cooling device operating a continuous rotary cycle. Through numerical analysis, we show that the cooling capacity and efficiency of the designed EC devices can be substantially enhanced by utilizing nanocomposites as the core refrigeration element. At a temperature span of 10 K, the refrigeration system using T-ND-2.6% as core components can achieve a coefficient of performance (COP) at 5.3 (0.8 when using the base terpolymer as core components) while maintaining a relatively high cooling power of over 240 W (a 10-fold cooling power improvement compared to the device based on the neat terpolymer.

## Results

### The overall performances

ND was selected owing to its high thermal conductivity, as we aimed to improve the heat transfer performance of polymeric nanocomposites. Besides the thermal conductivity, we anticipated that the introduction of low-k ND ($\varepsilon$ = 5 ~ 6)[50,51] into P(VDF-TrFE-CFE) would also enhance the interfacial polarization, thereby improving the ECE. As schematically shown in Fig. 1a, the permittivity of the ND was merely 1/8 of the polymeric matrix ($\varepsilon$ ~ 45) at room temperature (RT) and hence would also cause the electric field distortion at the heterogeneous interface of composites as what their high-k counterparts do[45]. Therefore, we hypothesized that the low-k fillers should enhance the ECE as well. Electron microscopic characterizations, e.g., transmission electron microscopy (TEM), and scanning electron microscope (SEM), indicated that the typical size of the filler is 5-10 nm as is shown in Fig. 1b. When the NDs are dispersed into the polymer, some of the NDs will inevitably agglomerate together due to size effect (Fig. 1c). The TEM results for T-ND-2.6% can be found in Supplementary Fig. 19.

The laser flash thermal conductivity testing instrument was utilized to measure the thermal diffusivity of the base terpolymer and nanocomposites. The thermal conductivity of different EC materials can be easily obtained from the thermal diffusivity. As depicted in Fig. 1d, the thermal conductivity of the EC materials exhibited an almost linear increase with the content of ND. The thermal conductivity of T-ND-1.2% and T-ND-2.6% was 35% and 75% higher than that of the base terpolymer, respectively. This trend is attributed to the large thermal conductivity of the ND particles to reduce the thermal resistance within the polymeric matrix, as schematically shown in Supplementary Fig. 2. The incorporation of 2.6 vol% of the ND resulted in an increase of thermal conductivity $\Delta\lambda$ of 0.15 W·m$^{-1}$·K$^{-1}$. The ratio $\Delta\lambda$/vol% is comparable to the best-reported values in the literature[44].

As naturally a great electrical insulator, ND-incorporation was observed to improve the electrical breakdown field of the EC nanocomposites, as evidenced by Fig. 1e. T-ND-2.6% exhibited a 25% enhancement in the electrical breakdown strength compared to that of the base terpolymer. We developed a phase-field model to evaluate the breakdown process of the base terpolymer and T-ND-2.6%. The results of the numerical modeling corroborated well with our experimental results (see Supplementary Section 2.2). Owing to the lower permittivity of the ND than that of the base terpolymer, the induced large

localized electric field would be shared on the side of the ND rather than the polymer, which is normally the case when one utilizes the high-k filler[52,53]. As the ND exhibits a higher dielectric strength (over 400 MV·m⁻¹) than the relaxor ferroelectric polymer, the ND-incorporation improves the electrical stability of the nanocomposites (Supplementary Fig. 4) as many other low-k fillers have shown[35,54].

Different from boron nitride nanosheet (BNNS), the other filler with high thermal conductivity, low-k and high dielectric strength that has been widely applied in EC polymer as fillers, the ND-incorporation observably enhanced the polarization of the base terpolymer rather than deteriorate the polarization as the BNNS does[35] (This is likely to indicate that the doping of BNNS reduces the ECE of the terpolymer[45], which can also be proved by our experiment in Supplementary Section 2.13). The EC-induced entropy changes ($\Delta S$) and temperature changes ($\Delta T$) of the base terpolymer and nanocomposites are presented in Fig. 1f, g, respectively. Detailed methods of measuring the EC-induced heat flux and temperature changes were elaborated in Supplementary Section 2.4. The cooling performance of the nanocomposites was greatly improved compared to that of the base terpolymer, i.e., at 100 MV·m⁻¹, a $\Delta S$ ~50.8 J·kg⁻¹·K⁻¹, corresponding to a $\Delta T$ ~10.7 K, was induced in T-ND-2.6%, which is about 60% higher than that of the base terpolymer ($\Delta S$ ~33.6 J·kg⁻¹·K⁻¹, corresponding to a $\Delta T$ ~ 6.7 K).

To validate the directly measured results of the EC-induced $\Delta S$, we further directly measured the $\Delta T$ via an infrared (IR) camera (see Supplementary Section 2.5). Owing to the heat loss that inevitably exists in the test condition, the IR measurement can only provide a result that is approaching the adiabatic temperature change (Supplementary Fig. 10). It is observed that the results of the temperature measurement are in agreement with those of the heat flux measurement. The IR images under 50 MV·m⁻¹ for nanocomposite films (T-ND-2.6%) with the electrode pattern (SJTU) are clearly shown in Fig. 1h, which is slightly lower than the result of the EC-induced $\Delta T$ deduced from the directly measured $\Delta S$ (Supplementary Fig. 11).

## Structural characterization

To understand the EC enhancement of the ND-incorporated EC nanocomposites, we explored the crystalline structural variation under the varying electric field. Small-Angle X-ray scattering (SAXS) and in-situ Wide-angle X-ray diffraction (WAXD) characterizations were employed to monitor the structural evolution of the base terpolymer and nanocomposites. As shown in Fig. 2a, after the ND-incorporation, the peaks for the non-polar phase (close to α phase in PVDF, see Supplementary Section 2.7) of the nanocomposites shifted to a lower diffraction angle, which is corresponding to a 0.02 Å

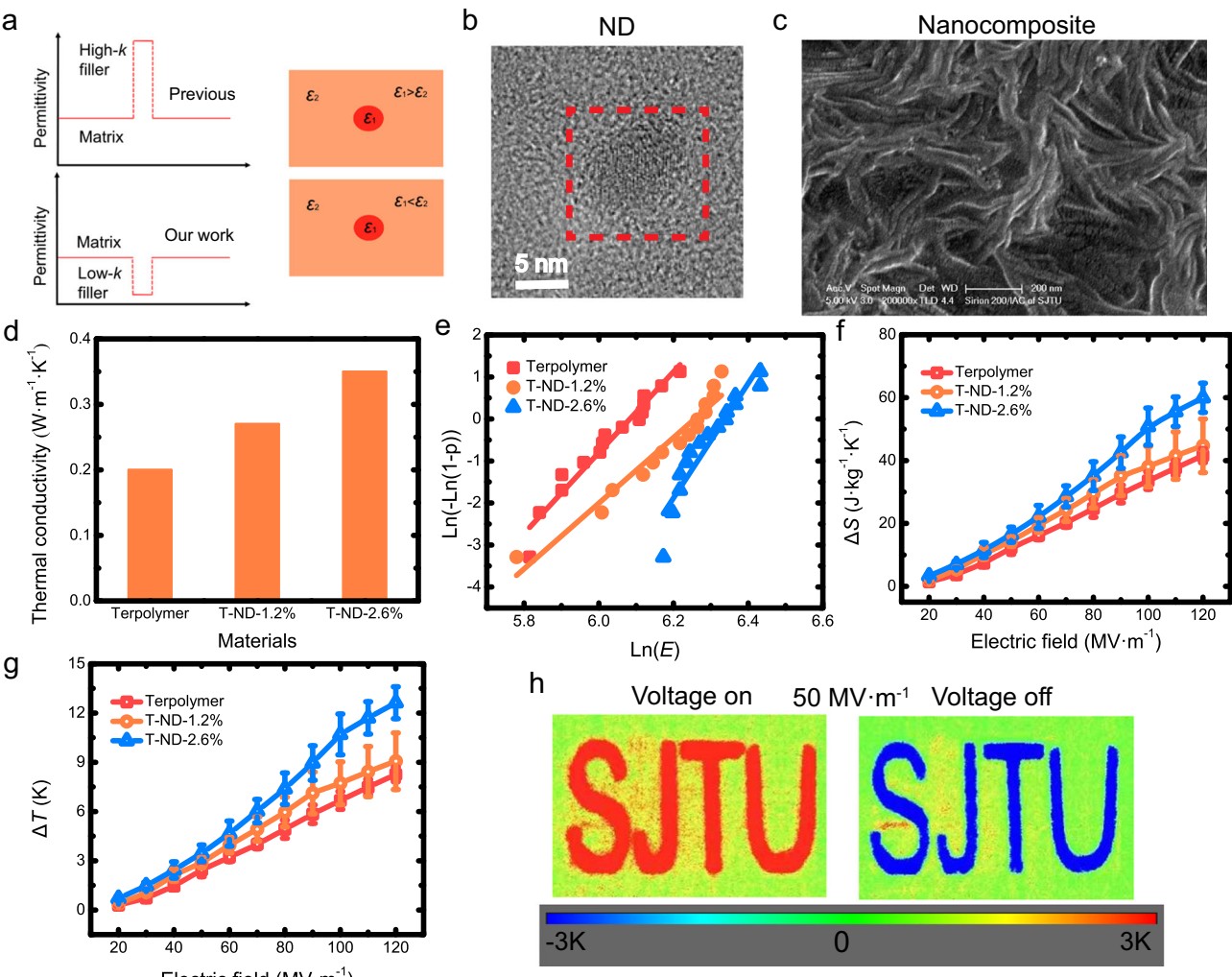

**Fig. 1 | Overall performance the base terpolymer and ND-incorporated nano-composites. a** Schematic representation of the filling strategy for composites. **b** TEM image of the ND. **c** SEM image of the nanocomposite (T-ND-2.6%). **d** Thermal conductivity of the base terpolymer and nanocomposites. **e** Weibull plots of the base terpolymer and nanocomposites. EC-induced entropy changes (**f**) and temperature changes (**g**) of the base terpolymer and nanocomposites as functions of the applied field amplitude at RT. Sample quantities $n \geq 3$, points are centered on the mean, and the bars indicate ±SD. **h** Infrared radiation temperature changes at 50 MV·m⁻¹ for nanocomposite films (T-ND-2.6%) with the electrode pattern.

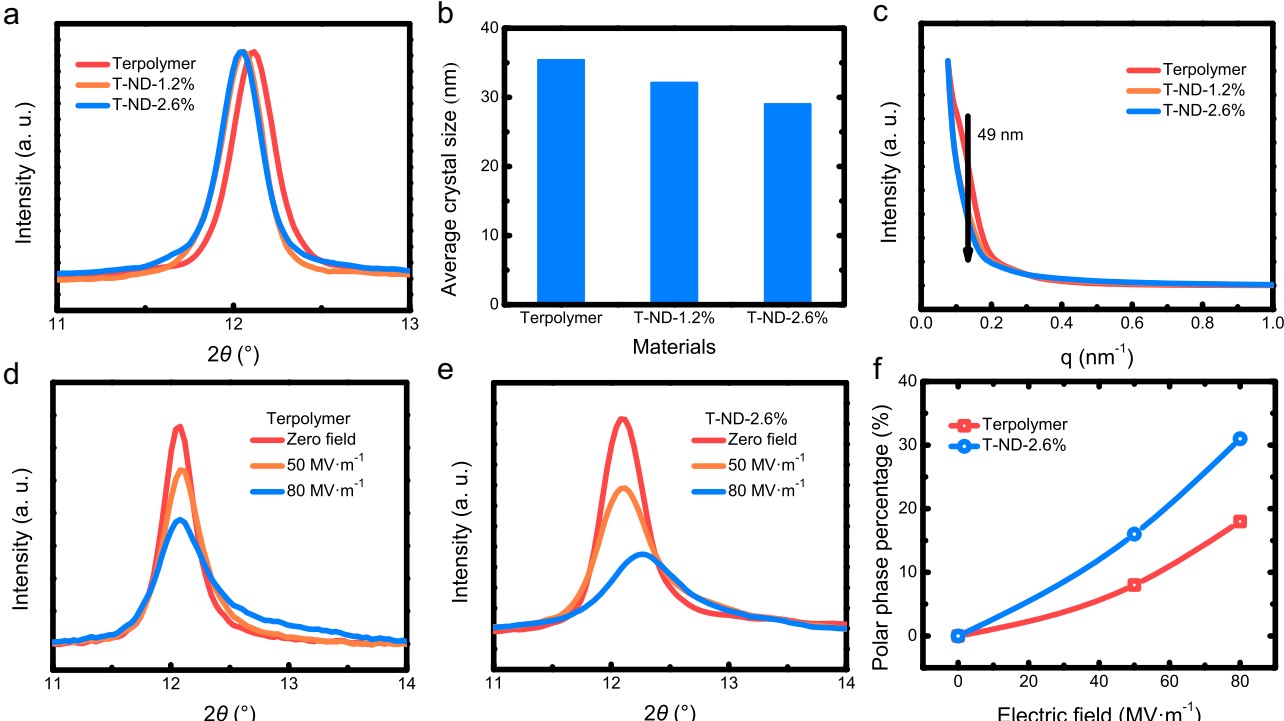

**Fig. 2 | SAXS and in-situ WAXD tests indicate the low-k ND could reduce the energy barrier of phase transitions in the terpolymer. a** WAXD spectra of the base terpolymer and nanocomposites. **b** Average crystal size of the base terpolymer and nanocomposites. **c** SAXS spectra of the base terpolymer and nanocomposites. In-situ WAXD spectra of the base terpolymer (**d**) and T-ND-2.6% (**e**). **f** Fraction of polar-phase changes as a function of the electric field.

expansion of the spacing of the crystal plane compared to that of the base terpolymer[55,56]. Our focus is on the distinctive Bragg peaks resulting from the two diffraction lines associated with the crystallographic planes of (200) and (110). This phenomenon can be correlated with the interplanar distance ratio, denoted as d200/d110, characterizing the orthorhombic pseudohexagonal phase, which is commonly identified as the relaxor ferroelectric feature[57–59]. In addition, the average crystal size in T-ND-2.6% was reduced to 29 nm from 35 nm of the base terpolymer, as shown in Fig. 2b. The decrease in the averaged size of crystalline structures in T-ND-2.6% indicates a significant rise in the number of crystallites and the presence of regions characterized by crystalline–amorphous interfaces. In addition, the reduced size of the crystalline structures could also be verified by differential scanning calorimetry (DSC) (Supplementary Fig. 12a, b). Although the melting enthalpies of the base terpolymer and nanocomposites showed no clear distinctions, the melting temperatures of the nanocomposites were observably lower than that of the base terpolymer (133.5 °C vs. 135.6 °C), which corroborates the results from the WAXD tests.

The extrinsic, small ND fillers introduce a dramatically increased area of the heterogeneous interface, which could strongly modulate the crystallization of the base terpolymer[60]. The expanded spacing of the crystalline plane and the reduction of the crystalline size would contribute to a more randomized polar structure that could enhance the ECE in the ND-incorporated nanocomposites. The interface between the organic and inorganic components may accommodate numerous defects, acting as nucleation sites to promote the creation of diminutive crystallites. These crystallites tend to be isolated and spatially localized, thereby impeding the long-range ordering within the polymeric matrix. SAXS data in Fig. 2c (also see Supplementary Section 2.7 and Supplementary Fig. 14) indicates crystal thinning along the chain direction, evidenced by a correlation peak representing the periodic organization of crystalline lamellae. The long period ($L_P$) can provide insights into the arrangement of crystalline lamellae and

amorphous regions. The reduction of the peak signal corresponding to the long cycle indicates that NDs may suppress the long-range ordering in the polymeric matrix.

$$L_P = \frac{2\pi}{q} \tag{1}$$

Achieving a substantial ECE at low fields demands an efficient transition from a high-polar-entropy state to a low-polar-entropy state[11]. We conducted the in-situ WAXD to study the field-induced structural change in the base terpolymer and nanocomposites (Fig. 2d, e), respectively. T-ND-2.6% presented a much stronger field-induced variation of the crystalline structure. Quantitatively, T-ND-2.6% showed that about 20 vol% of the non-polar phase transformed into the polar phase when the electric field was increased to 50 MV·m⁻¹, whereas for the base terpolymer only 8 vol% was converted (Fig. 2f and Supplementary Table 1). Hence, the low-k ND could reduce the energy barrier of phase transitions in the terpolymer, which would work with the enhanced polar randomness to further enhance the ECE.

## Dielectric analysis

To look into the correlation between the modified crystalline structure in the ND-incorporated nanocomposites and the enhanced ECE, we further studied the dielectric properties of the nanocomposites. According to the Landau–Devonshire phenomenological theory, the relationship between the change in entropy of EC materials and the change in polarization is described below[61–63],

$$\Delta S = -\frac{1}{2}\beta(P^2(E_h) - P^2(E_l)) \tag{2}$$

where $E_l$ is the low electric field and $E_h$ is the high electric field. Temperature-dependent permittivity and the polarization-electric field (P-E) loops were characterized, respectively. Therefore, the $\beta$

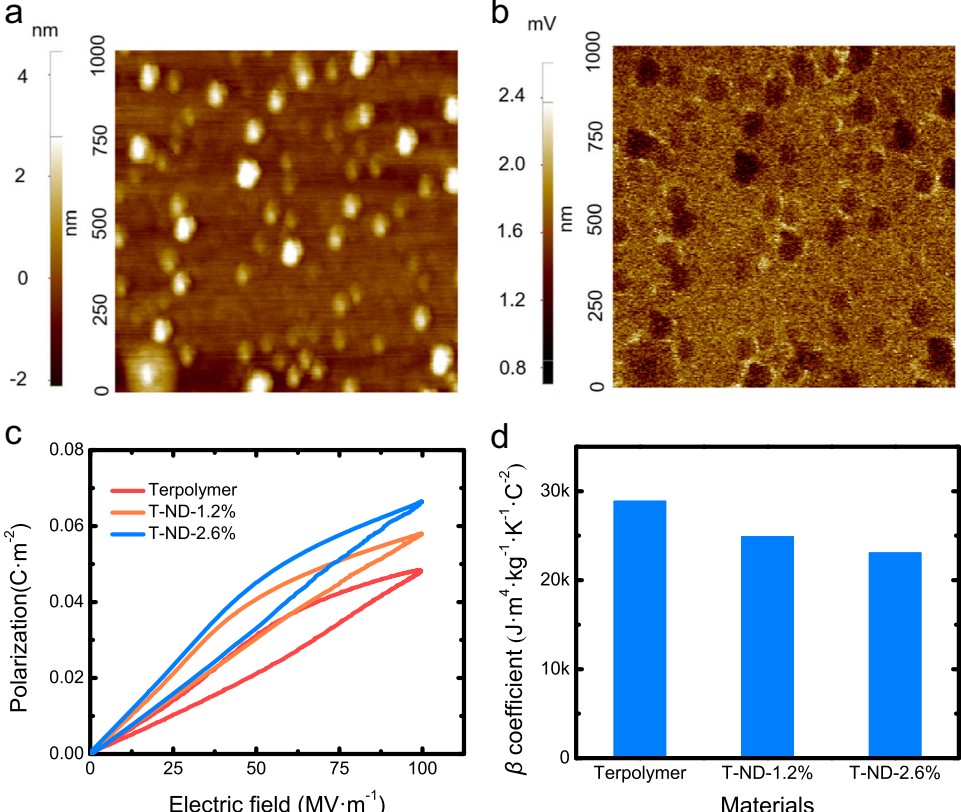

**Fig. 3 | Characterization of the main causes of polarization enhancement.**
**a** Topography signal near embedded ND. **b** Potential signal near embedded ND.
**c** Corrected P–E loops for the base terpolymer and nanocomposites at 10 Hz and
RT, with a maximum field of 100 MV·m⁻¹. **d** Ratio of the $\beta$ coefficient of the base
terpolymer and nanocomposites.

coefficient is essentially describing how efficient the maximum polarization is in generating the ECE. Owing to the low-k nature and the low content of ND, we observed no significant enhancement in the permittivity of the nanocomposites near RT (Supplementary Fig. 15) and a much-enhanced polarization under high electric fields, albeit the reduced crystalline size. The phenomenon indicates the ND-incorporation does not change much of the polar correlation. The above analysis suggests that the origin of the consequently enhanced ECE is different from the polar high entropy polymers[11]. Therefore, the enhanced ECE may be attributed to the increased polarization that was induced at the internal interfaces.

To visually verify the existence of the interfacial polarization, we employed EFM to evaluate the interfacial coupling effect of the ND-incorporated nanocomposites (See Supplementary Section 2.10 for details). The surface morphology and potential distribution of the nanocomposites are shown in Fig. 3a, b, respectively. Owing to the low-k, electric potential at the locations of ND was in average lower than that at the polymeric matrix. However, a significant build-up of potential can be observed at the interface between the polymeric matrix and inorganic fillers (Fig. 3b). In principle, the dipolar polarization can largely affect the surface potential of ferroelectric polymers[46]. Therefore, the microscopic images suggest the origin of the enhanced polarization after the incorporation of the low-k fillers.

On the other hand, the dielectric losses of the nanocomposites were greatly enlarged at temperatures slightly above the RT, which is consistent with the P–E loops at RT (Supplementary Fig. 17). The increased dielectric loss may be attributed to the random defects induced by the ND-polymer interface, which facilitates the hopping of the charged carriers. We observed that the conduction loss was increasing with the increasing content of ND, which may cause errors

when we study the contribution of the orientation polarization that directly contributes to the ECE.

To single out the orientation polarization, we subtracted the conduction loss to correctly reveal the polarization-entropy correlation[64]. The conduction loss from leakage current contributes to the nominal polarization that was measured, so we subtracted the contribution from the measured nominal polarization to obtain the real polarization and the correction method can be found in Supplementary Section 2.8. The corrected P–E loops of the base terpolymer and nanocomposites under 100 MV·m⁻¹ at 10 Hz and RT are shown in Fig. 3c. The maximum polarization was significantly increased in the nanocomposites. T-ND-2.6% exhibited an induced polarization of 0.066 C·m⁻² under 100 MV·m⁻¹, which was about 37.5% higher than the base terpolymer (0.048 C·m⁻²). Combing the reduction of the energy barrier and the enhanced interfacial polarization[47], the ND-incorporated nanocomposite exhibited the modified field-induced entropy changes compared to the base terpolymer.

In ferroelectrics and dielectrics, not all polarization could contribute to the ECE[60]. The effectiveness of polarization-induced entropy changes can vary over a great range[11]. To determine how effective the interfacial polarization induces ECE, we studied the $\beta$ coefficient in the Landau–Devonshire phenomenological theory (Eq. (2)). According to Fig. 3d, the $\beta$ coefficient of the ND-incorporated nanocomposites was slightly smaller than that of the base terpolymer. Although the overall ECE was enhanced, the reduced $\beta$ coefficient indicated that the interfacial polarization is not as effective as the base terpolymer in terms of generating polar entropy changes. The polar inactive carbon material may restrain the polymer chain mobility under the electric field[65]. It is important to note that significantly enhancing the local electric field could overcome the reduced $\beta$ coefficient and improve the overall ECE.

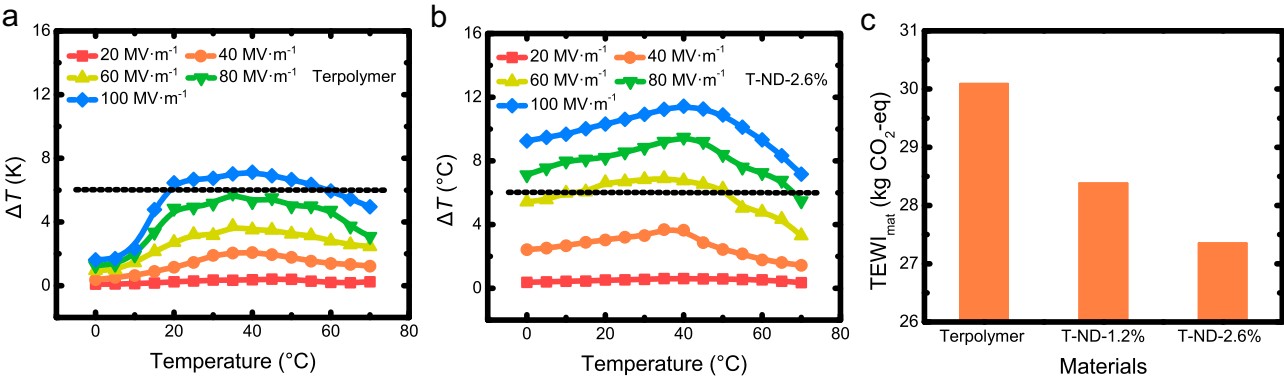

**Fig. 4 | Environmental friendliness of the ND-incorporated nanocomposites. a, b** Temperature-dependent EC-induced temperature changes of the base terpolymer and T-ND-2.6%. **c** TEWI of the base terpolymer and nanocomposites.

In addition, we compared the volumetric enhancement of ECE (the ratio between the EC enhancement in percentage and the volume percentage of the filler) at 100 MV·m$^{-1}$ and RT, for different nano-fillers. The ND introduced the highest volumetric EC enhancement (~23%/vol %) when compared with other fillers in the previous study[45,47,48] (see Supplementary Fig. 8).

In addition, we note that the low-k ND incorporation could efficiently enhance the temperature independency of the ECE. For conveniently designing an EC refrigeration device, its working body is expected to feature a temperature-invariant cooling capability across a broad temperature range. The EC-induced temperature changes of the base terpolymer and T-ND-2.6% were tested from 0 to 70 °C, as shown in Fig. 4a, b. Under 100 MV·m$^{-1}$, T-ND-2.6% can reach an EC-induced temperature change of more than 6 K in a temperature range of over 70 K, while the corresponding temperature range for the base terpolymer was approximately 40 K.

The improved ECE, EC strength, and temperature independency in the T-ND-2.6% could lower the overall carbon emission of a potential EC device. To quantify the emission reduction, we studied the material-specific total equivalent warming impact (TEWI$_{mat}$) of the working body under a fixed device working condition. The TEWI$_{mat}$ is a measure of the greenhouse gas emissions associated with a particular product, including its activity, or process, expressed in terms of its impact on global warming[20]. It takes into account both direct emissions (if a refrigerant with a finite GWP was used), as well as indirect emissions (electricity consumption) associated with the energy used to operate the product, activity, or process.

TEWI is a commonly-used metric for evaluating energy-consuming technologies with $CO_2$-associated origins. In conventional vapor compression refrigeration systems, TEWI comprises two key components: (1) direct emissions of high GWP refrigerant vapors and (2) electricity consumption, quantified as an equivalent $CO_2$ emission, contributing indirectly to TEWI.

$$TEWI = CO_{2,direct} + CO_{2,indirect} \quad (3)$$

where $CO_{2,\,direct}$, $CO_{2,\,indirect}$ are the direct and indirect equivalent $CO_2$ emission.

$$CO_{2,direct} = x * GWP \quad (4)$$

$$CO_{2,indirect} = \frac{Q}{COP} * T_{CO_2} * L \quad (5)$$

where $x$ denotes the weight of vapor emissions, $Q$ represents the cooling capacity of the refrigeration system, $T_{CO_2}$ signifies the average

mass of $CO_2$ released per kWh of electrical energy in power plants, and $L$ stands for the lifetime (in years) of the refrigeration system.

TEWI$_{mat}$ is closely related to the material COP of the EC material, as detailed by Qian et al.[14]. Specifically for caloric cooling, COP$_{mat}$ can be represented as:

$$COP_{mat} = \frac{Q}{W_{net}} = \frac{T_C \Delta S - \frac{1}{2} Hy}{(T_H - T_C)\Delta S + Hy} \quad (6)$$

where $W_{net}$ represents the net input work per unit mass, $T_C$ and $T_H$ denote the temperatures at the cold and hot ends, respectively, and $Hy$ stands for the hysteresis in caloric materials.

Naturally, as a solid-state refrigerant, the TEWI of the EC material has no direct emission part (Fig. 4c)[66]. By fixing the working condition of a potential EC cooling device, we could use the TEWI to evaluate any EC material that can fit into the working condition[14]. The major input to the TEWI of the EC materials is the hysteresis and conduction loss of the materials during the electrical cycling. By utilizing the directly measured P-E loops, we can conclude that the TEWI of the ND-incorporated nanocomposites was significantly lower compared to the base terpolymer (from 30.1 kgCO$_2$-eq to 27.4 kgCO$_2$-eq). The concurrently enhanced ECE, EC strength, and reduced dielectric loss demonstrate the great potential of the ND-incorporated EC nanocomposites for a sustainable future of refrigeration.

## System cooling performance evaluation

The performance of EC materials can be assessed through their application as core components in refrigeration devices. By analyzing the cooling power and efficiency of EC devices utilizing various materials, one can identify the most suitable EC material for a given refrigeration system. In this study, we proposed a design concept for a rotary EC device that incorporated fluid-solid conjugated heat transfer to meet the demand for highly efficient and compact cooling where the size and weight had high priority. The structure of the designed EC device is shown in Fig. 5a and numerical simulation was conducted to further evaluate its performance.

The simulation parameters were obtained based on the experimental results mentioned previously. Referring to the previous study[67], deionized water was selected as the heat transfer fluid. The detailed working principles, system structures, initial and boundary conditions, and other information are presented in Supplementary Section 3. The primary objective of improving the thermal conductivity of EC materials is to enhance the heat exchange between EC materials and heat transfer fluid.

The hot fluid outlet is designated as "face 1," while the cold fluid outlet is termed "face 2" (fluid flow occurs only during the fluid flushing

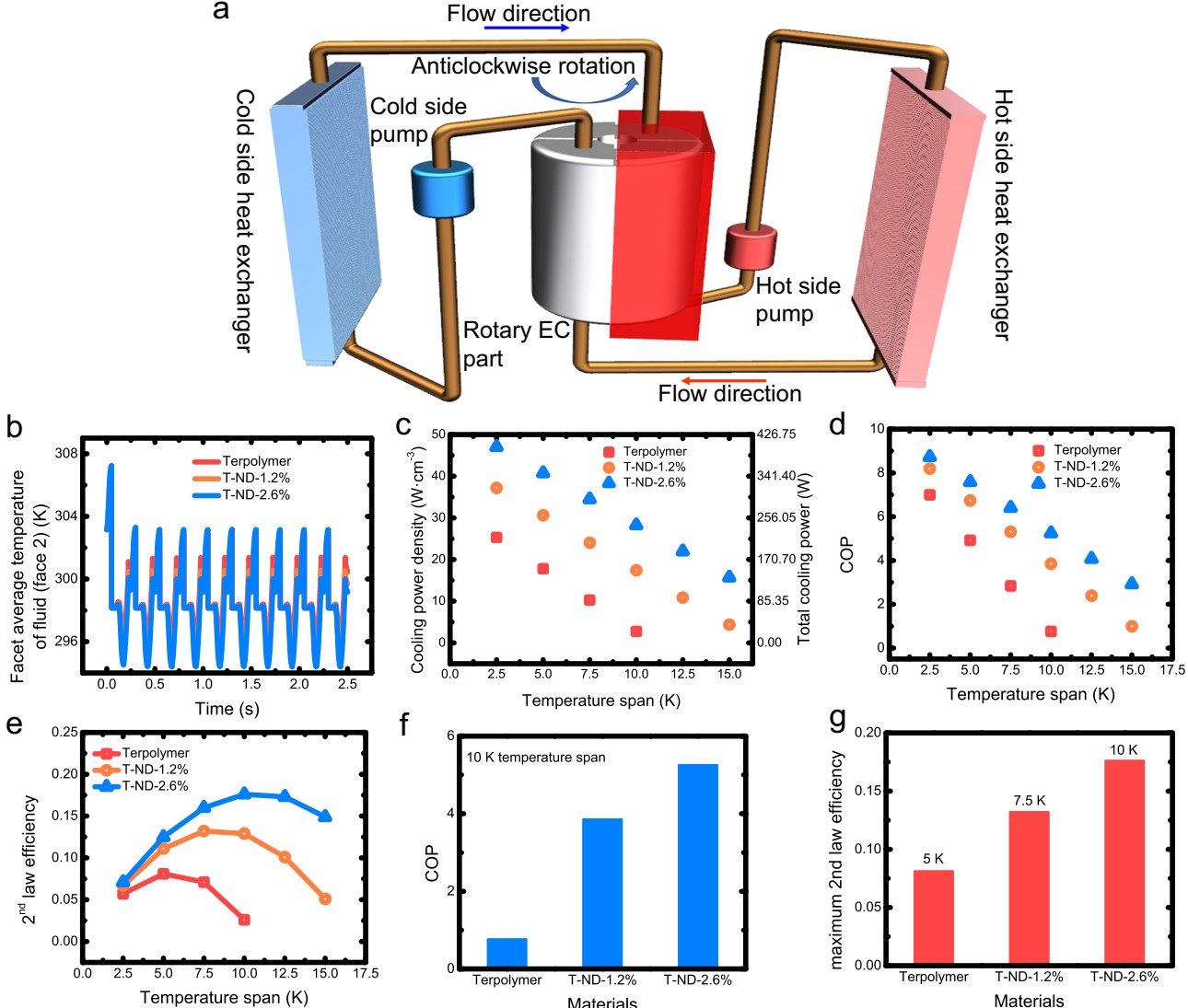

**Fig. 5 | Cooling performance of the fluid-solid coupling-based EC refrigerator. a** Schematic diagram of the layout of the EC refrigeration system. **b** Temporal resolutions of facet average temperatures of fluid (face 2). **c** Cooling capacity corresponding to the use of three types of EC materials over different temperature spans. COP (**d**) and 2nd law efficiency (**e**) corresponding to the use of terpolymer and nanocomposites over different temperature spans. **f** COP of refrigeration devices based on different EC materials at a temperature span of 10 K. **g** Maximum 2nd law efficiency of refrigeration devices using different EC materials and working temperature spans.

stage, as observed in Supplementary Figs. 29 and 31). Once the system achieves stability, the deionized water temperature within the EC part and at the outlet cyclically varies with operating time. Figure 5b, c present the temporal resolutions of average fluid temperatures on face 2 and the total cooling capacity, based on different EC materials.

As shown in Fig. 5c, the optimal performance in terms of total cooling power and cooling power density (CPD) for the EC device is achieved when T-ND-2.6% is utilized as core components. This can be attributed to the exceptional cooling and heat transfer capabilities exhibited by T-ND-2.6% compared to the other two types of EC materials. At a temperature span of 10 K, the system can achieve a CPD of 28.24 W·cm$^{-3}$ and total cooling power of 241.2 W with the utilization of T-ND-2.6%. Compared to the refrigeration system based on neat terpolymer, it offered more than a 10-fold cooling power improvement.

The comparative cooling efficiency of the EC device based on various materials is clearly shown in Fig. 5d–f. Consistent with the trend observed for CPD, the system utilizing T-ND-2.6% as core components exhibited the highest coefficient of performance (COP),

followed by T-ND-1.2%. These findings further underscore the notable benefits of incorporating the nanocomposites as core components in the designed EC device. When operating at a temperature span of 10 K, the refrigeration system utilizing T-ND-2.6% as core components exhibited a COP of 5.3, which is more than 6 times higher than that of the EC device based on the neat terpolymer. These results suggest that the incorporation of nanocomposites with superior heat transfer and cooling performance can significantly advance the research and application of high-power EC devices.

We can also determine the best working temperature span of certain EC materials based on the variation trend of 2nd law efficiency. As depicted in Fig. 5g, the EC device utilizing T-ND-2.6% as core components can achieve the maximum 2nd law efficiency (17.6%) when operating at a temperature span of 10 K. In contrast, the efficiency of the refrigeration system based on the neat terpolymer is merely 2.6% under the same conditions. In other words, in terms of cooling efficiency, terpolymer-based cooling systems are more suitable for operation at a 5 K temperature span. This is also a good example of the advantages of the ND-incorporated nanocomposites, i.e., systems

based on the nanocomposites can achieve better cooling efficiency and operate over a wider temperature span.

In addition, we developed a film-like EC oscillating refrigerator that is in a similar configuration and operating strategy as the previously reported EC device[27] (see Supplementary Section 4). The experimental results of this platform also illustrated the superior cooling capacity (the same CPD can be obtained at lower electric fields) and the better heat transfer performance (shorter heat dissipation time, see Supplementary Fig 34) of the ND-incorporated nanocomposites compared to the base terpolymer.

## Discussion

In this paper, we demonstrated that by counter-intuitively including a non-polar, low-k carbon-based nano-insulator, the ferroelectric-based polymeric composites exhibited a markedly improved polarization and ECE. We observed that the low-k ND could also induce a large localized field at the organic-inorganic interface that may contribute to the enhanced polarization and ECE, similar to their high-k, ferroelectric counterparts. In addition, the dielectric strength could also be improved owing to the insulative nature of the ND. By multiple structural characterizations, we demonstrated that the ND could expand the spacing of the crystal plane, reduce the crystal size of the polymer matrix, and reduce the energy barrier between the polar and nonpolar phases. Compared to the base terpolymer, the ND-incorporated nanocomposite exhibits an improved TEWI and temperature independency, which are key requirements for designing an EC cooling device. Furthermore, the thermal conductivity could also be slightly improved as a cherry on top of the sundae. The overall improvement of the ND-incorporated relaxor ferroelectric polymer could significantly improve the COP and CPD of the EC device, utilizing the fluid-solid conjugate heat transfer. The demonstration of the EC enhancement by the low-k ND could broaden the pool of the filler for searching EC nanocomposites for sustainable refrigeration in the future.

## Methods

### Preparation of polymer and ND-incorporated nanocomposites
First, ND was dispersed in DMF for 6 h (using an ultrasonic cell pulverizer) and an ice bath environment was used to keep the mixture temperature constant. The ND particles was homogeneously mixed with a solution of P(VDF-TrFE-CFE) in DMF. The resulting mixture was cast onto a clean glass slide and dried at 60 °C for 8 h. Following the evaporation of the solvent, the composite films were carefully peeled off and subjected to annealing at 115 °C for 12 h to enhance crystallinity and eliminate the remaining residual solvent. As a comparison, the base polymer films were made using the same solution-casting method. For dielectric measurements, gold electrodes were sputtered on both surfaces of the polymer films. More detailed information on the process of film preparation can be found in Supplementary Section 1.1.

### Thermal conductivity measurement
The thermal conductivities of the base polymer and nanocomposites were measured using a laser flash thermal conductivity testing instrument (LFA467 HyperFlash, Netzsch). The device can be used to test the thermal diffusivity of EC materials. The thermal conductivity ($\lambda$) was calculated from $\lambda = \rho$ (density, kg·m$^{-3}$) $\times C_p$ (specific heat, J·kg$^{-1}$·K$^{-1}$) $\times D$ (thermal diffusivity, mm$^2$·s$^{-1}$). The thermal conductive specimen diameter and thickness were 12.7 mm and ∼0.1 mm, respectively. The specific heat capacities of the base polymer and nanocomposites were measured via the DSC 25 from TA Instruments. The three-step test was adopted to obtain the specific heat of the EC materials. The testing method and results for specific heat can be found in Supplementary Section 2.1.

### ECE measurement
ECE measurements were performed in an in-situ calibrated calorimeter, using a heat flux sensor (27134-1, RdF Corporation). High voltage was delivered via a Trek 610C high-voltage amplifier, and the signal for generating the waveforms was generated by an arbitrary function generator (Stanford Research Systems Model DS345). In addition, the calorimetric results were cross-checked using an infrared camera (FLIR A655sc) positioned 30 cm above the EC film. More details about the measurement of ECE can be found in Supplementary Sections 2.4 and 2.5.

### WAXD and SAXS measurements
WAXD and SAXS were conducted at the BL19U2 beamline of Shanghai Synchrotron Radiation Facility (SSRF), utilizing X-rays with a wavelength of 1.03 Å. The data acquisition time for each frame in both WAXD and SAXS was controlled at 15 s. The sample-to-detector distances were 207 mm for WAXD test and 2030 mm for SAXS test. Additional details are available in Supplementary Section 2.7.

### Ferroelectric and dielectric testing of EC materials
Permittivity spectroscopy were assessed using a precision LCR meter (HP 4284A) equipped with a temperature chamber and polarization-electric field loops were measured through a modified Sawyer-Tower circuit. In addition, dielectric strength was tested by Shanghai Juter High Voltage Electrical & Equipment Co. under a direct current (d.c.) voltage at a rate of 200 V·s$^{-1}$.

### DSC
The latent heat during the phase transitions of the terpolymer and ND-incorporated ones was measured by utilizing the Differential Scanning Calorimetry instrument (DSC25, TA Instruments). The experimental procedure started with an initial cooling step to −50 °C, followed by the cyclic ramping between −50 °C and 160 °C at a rate of 20 °C·min$^{-1}$. The mass of EC materials was controlled at 10 mg.

### TEM
The microstructures of ND were characterized by field emission transmission electron microscope (FE-TEM, Talos F200X G2, Thermo, USA).

### SEM
The microstructures of the ND, terpolymer and nanocomposites were characterized by field emission scanning electron microscopy (FE-SEM, Nova NanoSEM 450, FEI, USA).

### EFM
The topography signal and potential signal near embedded ND were characterized by Electrostatic Force Microscopy (EFM, Park NX10). More details can be found in Supplementary Section 2.10.

### Reporting summary
Further information on research design is available in the Nature Portfolio Reporting Summary linked to this article.

## Data availability
All data supporting the findings of this study are available within the article and in the Supplementary Information. Other relevant data are available from the corresponding author X.Q. (xsqian@sjtu.edu.cn) on reasonable request. Source data are provided with this paper.

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

## Acknowledgements

This work was supported by the National Key R&D Program of China (2020YFA0711500), and the National Natural Science Foundation of China (52076127 and 51776119), the Natural Science Foundation of Shanghai (20ZR1471700, 22JC1401800), the State Key Laboratory of Mechanical System and Vibration (Grant No. MSVZD202211), the Oceanic Interdisciplinary Program of Shanghai Jiao Tong University (project number SL2020MS009), the Prospective Research Program at Shanghai Jiao Tong University (19×160010008), the Changzhou Leading Talents Project, and the Student Innovation Center and the Instrumental Analysis Center at Shanghai Jiao Tong University. We acknowledge the support by Shanghai Jiao Tong University 2030 Initiative. We thank N. Li from the BL19U2 beamline and X. Miu from the BL16B1 beamline of Shanghai Synchrotron Radiation Facility for help with synchrotron X-ray measurements.

## Author contributions

X.Q. conceived the concept. X.Q. and Q.L. designed the experiment and wrote the manuscript. Q.L., L.W., N.Z., D.H., F.D., and S. Zheng carried out the material preparation and characterization. L.W., and N.Z. conducted the EFM tests. X.Q., Q.L. and D.H. carried out the synchrotron X-ray measurements. H.H. and X.S. carried out the phase-field simulation. Q.L. and J.S. designed the model for cooling devices. J.S. and J.C. supervised the device modeling. X.Q. and C.D. supervised the project. All authors analyzed and interpreted the data.

## Competing interests

The authors declare no competing interests.
