## [Peer Review File · Nature Communications]

REVIEWER COMMENTS

Reviewer #1 (Remarks to the Author):

In this manuscript, the authors adopt nanodiamond (ND) as low-k and extrinsic dielectric fillers to fabricate polymer nanocomposites for EC refrigeration. The ND markedly enhanced the polar activity and electrocaloric effect (ECE) in relaxor ferroelectrics. A low content of 2.6 vol% resulted in a polymer nanocomposite that exhibited an overall improvement of material properties. The research results are interesting for the readers to a certain extent. However, similar studies on improving the electrocaloric performances of ferroelectric polymer by means of composite materials have been widely reported (e.g. *Polymer* 2013, 54, 5299; *ACS Applied Energy Materials* 2018,1, 1344; *Advanced Materials* 2019, 31, 1801949; *Composites Part B: Engineering* 2021, 227, 109391). In brief, this work is not innovative enough to be published in the *Nature Communication*.

1. Low-k composites used to improve the electrocaloric performance have been reported in previous years (e.g. *Polymer* 2013, 54, 5299). In the title, the authors proposed that low-k nano-dielectrics modulated phase transition to improve the electric refrigeration effect. From the results, it could be seen that the composite of diamond had little effect on the phase transition temperature of the polymeric relaxor ferroelectrics, which could be confirmed in the dielectric temperature spectrum (Supplementary Fig. 13). The title should clearly express the focus of the article.

2. Compared with other literature (Ref. 47: *Adv. Mater.* 2019, 31, 1801949, at $100 \text{ MV}\cdot\text{m}^{-1}$, $\Delta T \sim 19.8 \text{ K}$, which is about 125% higher than that of the terpolymer ($\Delta T \sim 8.8\text{K}$)), this work was not as excellent as the authors' claim ("These enhancements are even more effective compared to their high-k counterparts, i.e., the inclusion of 2.6 vol% of ND ... could increase the ECE for 60% at an electric field of $100 \text{ MV}\cdot\text{m}^{-1}$... The value is nearly 70% higher than the best-reported high-k fillers⁴⁷"). Then, the ND nanocomposite had an obvious current leakage and conduction loss at high field, which was extremely unfavorable for the practical device applications.

3. In Fig. 2f, when the electric field was zero, the proportion of the polar phase to 0 % was not rigorous. Undetectable by XRD did not mean that the phase was not present.

4. The particle size of nano-diamond affected the interface area and polarization. From the results of AFM and SEM, the diameter distribution of the ND was about 50-100 nm, which was not consistent with the 10 nm mentioned by the authors.

5. Compared with the terpolymer matrix, the melting enthalpy of the nanocomposite film decreased, but the crystallinity in Supplementary Table 1 increased. And as the electric field increased, the grain size of the polar phase increased first and then decreased. Was there a problem with the fitting method used?

6. In page 9, the authors proposed that "The thermal conductivity is attributed to the large thermal conductivity of the ND particles to reduce the thermal resistance within the polymeric matrix."

However, the effect was not significant. Why were the excellent characteristics of high thermal conductivity fillers not reflected? In addition to the percolation theory, what were the possible reasons? If increasing the proportion of the nano-diamond composite could improve the thermal conductivity, what effect would it have on the electrocaloric performance of the system?

7. In page 10, the authors cited the literature on boron nitride for comparison (Literature 54), and the cited article is incorrect. The authors stated that “Different from boron nitride nanosheet (BNNS), the other filler with high thermal conductivity, low-k and high dielectric strength that has been widely applied in EC polymer as fillers, the ND-incorporation observably enhanced the ECE of the base terpolymer rather than deteriorate the ECE as the BNNS does⁵⁴”. Literature 54 does not study the impact of the BNNS on the ECE, much less the deteriorate described by the authors.

8. In page 15, the authors mentioned that “To single out the orientation polarization, we subtracted the space charge-induced polarization to correctly reveal the polarization-entropy correlation”. How to accurately calculate the space charge-induced polarization?

9. In the Landau–Devonshire phenomenological theory (equation. (1)), was it applicable to both bulk and composite materials? If applicable, how to clearly distinguish dipole polarization from space charge-induced polarization in this formula? Did the electrocaloric effect necessarily come from the dipole steering polarization and the interface effect?

10. The authors demonstrated the advantages of the composite materials only through the simulation of the fluid-solid coupling-based EC refrigerator, lacking experimental results of real devices. Then, both the device structure and the heat transfer simulations in SI have been published in the authors' earlier work (Applied Thermal Engineering 2021, 190, 116806).

11. The notes in the data chart were inconsistent with the text, please check it carefully (Fig. 5 T-TD-2.5%, T-TD-5%?). The annotation information of the sample loss value in Supplementary Fig.13d was wrong.

Reviewer #2 (Remarks to the Author):

Authors reported the well-studied P(VDF-TrFE-CFE), and introduced a low content of nanodiamond (low dielectric constant) to improve the overall performance of the EC polymers. It's interesting that nanodiamond can promote the interface polarization and reduce the phase transition energy barrier due to the significantly different dielectric constant from the polymeric matrix. The entropy change of T-ND-2.6% is improved by 60% as compared to the already existed high entropy change in the base terpolymer. In addition, the thermal conductivity and dielectric insulation of the ND-incorporated nanocomposites have also been improved to varying degrees. Overall, the paper is well written with some interesting findings, and the comparison based on the current results and reported literatures are performed. However, some comments/questions need to be amended prior to the final acceptance. Detailed comments are included as below.

(1) The direct method is a widely used technique for quantifying the EC-induced temperature changes. As mentioned in the manuscript that the temperature change of the EC materials is obtained by using the infrared camera, showing the slightly smaller value than that of obtained by the direct method. Can authors please provide the explanation? Moreover, how to ensure the accuracy of the infrared camera test?

(2) Adding some defects in the base terpolymer helps the crystallinity, which is also observed in this work and this crystallinity of course improves the EC performances. My question is what is the specific contribution of crystallinity when compared to the contribution of polar/non-polar transition? Further analysis is required.

(3) Compared to other refrigeration technologies, the TEWI values of EC materials are significantly lower. For instance, the improved ECE, EC strength, and temperature independency in the T-ND-2.6% could lower the overall CO₂ emission of a potential EC device. It would be useful for the authors to give how the TEW_{mat} is calculated so that this reported results can be more easily understood for a wider audience.

(4) In this work, numerical simulations were carried out on the designed rotary EC device. The authors claim to have performed a grid-independent verification in the Supplementary Section 3.2. It is recommended to provide the detailed results and discussion of the grid independence verification to guarantee the accuracy of the simulation.

Reviewer #3 (Remarks to the Author):

The authors report on the improved electrocaloric properties of the ferroelectric terpolymer-based nanocomposites with nanodiamond fillers. These fillers with low dielectric constant not only somehow cause interfacial polarization and boost electrocaloric effect but also help to increase the thermal conductivity of the terpolymer to result in better cooling performance in cooling devices.

While the study presents some novel, interesting and counterintuitive results, several points need to be clarified before a decision can be made. Most importantly, the language of the manuscript is not clear therefore important messages cannot be conveyed enough. Significant improvement of the discussion and the writing is necessary.

1)The manuscript has some significant overlap with ref 11. The authors are suggested address this more and stress the differences and similarities more clearly. They state that the number of polar entities do not change or even decrease in the current manuscript unlike what was reported in ref. 11. Instead, they attribute the increased ECE to the increased polarization induced at the interfaces. However, similar to ref.11, here, the authors also suggest that reduced energy barriers for field-

induced non-polar to polar transition. Is this not playing role in the increased ECE? In another section, this is also held responsible. In other words: The origin of the enhanced electrocaloric effect is unclear from the discussion. Is it the field-induced non-polar polar phase transition, which is facilitated by the ND fillers, or is it the interface effect? The current discussion is confusing.

Also, the title of the manuscript can be more exact reflecting this facilitated field-induced transition: What the nanodiamond filler facilitates or modulates is the electric-field induced phase transition.

2) In Fig. 2a the authors show WAXD spectra of the terpolymer & nanocomposites and state that after ND incorporation, 'the peaks for the non-polar phase of the nanocomposites shifted to a lower angle.' They do not specify the non-polar or polar phases in general in the manuscript. Is the non-polar one α , polar one β ? More information is necessary about this section.

3) It is indeed surprising that low-k filler results in a similar interface effect as the high-k filler. The interface that forms between high-k filler and the polymer typically amplifies the external electric field due to the contribution from the easily polarizable high-k filler. How do the authors explain the observation of a similar interface effect with the low-k filler? How significant is the built-up potential at the interface shown in Fig. 3a-b. Can the authors compare this with the same potential reported at the interface between the polymer and high-k filler?

4) It is quite difficult to conclude from the SEM image in Fig. 1c that the dispersion of the nanoparticles in the polymer matrix is homogeneous. The nanoparticles are very small but still...

5) On line 157, it is stated that '...electrical breakdown field of EC nanocomposites, as evidenced by Fig. 1d'. Figure number is confused apparently.

6) The authors do not describe what is shown in Fig. 2c adequately. How can the reader agree with their conclusion that long-range ordering is suppressed if they don't describe what they observe in the Figure for those who are not familiar with SAXS? This is a general problem. The way the authors write the manuscript does not help the reader enough to understand what the figures tell or how did the authors draw conclusions from the data in the figures.

7) Another example to the above problem: What is the β coefficient, what does it represent? Not clearly written.

Response to the Reviewers' Comments

Reviewer #1 (Remarks to the Author):

In this manuscript, the authors adopt nanodiamond (ND) as low-k and extrinsic dielectric fillers to fabricate polymer nanocomposites for EC refrigeration. The ND markedly enhanced the polar activity and electrocaloric effect (ECE) in relaxor ferroelectrics. A low content of 2.6 vol% resulted in a polymer nanocomposite that exhibited an overall improvement of material properties. The research results are interesting for the readers to a certain extent. However, similar studies on improving the electrocaloric performances of ferroelectric polymer by means of composite materials have been widely reported (e.g. Polymer 2013, 54, 5299; ACS Applied Energy Materials 2018,1, 1344; Advanced Materials 2019, 31, 1801949; Composites Part B: Engineering 2021, 227, 109391). In brief, this work is not innovative enough to be published in the Nature Communication.

Response: We thank the reviewer for the valuable comments on our work, which would help us to improve the current manuscript. Previous studies on EC composites have mostly been on the introduction of high-k fillers into the base terpolymer (P(VDF-TrFE-CFE))^{1, 2, 3, 4}. There is only one study of a low-k filler (Polymer 2013, 54, 5299), which used ZrO₂ nanoparticles with a dielectric constant ($\epsilon > 30$) close to that of the base terpolymer ($\epsilon \sim 45$). The focus of our study, however, is that incorporating a low content of low-k fillers (with a permittivity of merely 1/8 of the polymeric matrix) can play a similar role of the high-k fillers in enhancing the electrocaloric effect of the base terpolymer. Considering many high-k ferroelectric ceramic fillers are also EC active, this work is interesting to us because it provides a model to solely study the interface effect on the ECE without worrying about the potential contribution of the EC-active fillers on the EC enhancement. The detailed, point-by-point responses could be found below.

1. *Low-k composites used to improve the electrocaloric performance have been reported in previous years (e.g. Polymer 2013, 54, 5299). In the title, the authors proposed that low-k nano-dielectrics modulated phase transition to improve the electric refrigeration effect. From the results, it could be seen that the composite of diamond had little effect on the phase transition temperature of the polymeric relaxor ferroelectrics, which could be confirmed in the dielectric temperature spectrum (Supplementary Fig. 13). The title should clearly express the focus of the article.*

Response: Thanks for the suggestions. Chen et al⁵ (Polymer 2013, 54, 5299) improved the polarization and ECE by means of preparing (P(VDF-TrFE-CFE))/ZrO₂ nanocomposites. The permittivity of ZrO₂ nanoparticles ($\epsilon \sim 30$) is low compared to many oxides but is comparable to that of base terpolymer ($\epsilon \sim 45$). The small difference between the permittivity is insignificant to induce large internal field distortion at the organic-inorganic interface. The nanodiamond (ND) nanoparticles used in this study are a class of fillers with a low dielectric constant (5.6 vs. 45) and high electrical insulating properties. With the large discrepancy in permittivity, the ND-incorporated nanocomposite exhibited a 60% EC enhancement compared to the 20% achieved by Chen et al. Besides the largely enhanced performance, this study is intended to demonstrate that the low-k fillers (ZrO₂ is not low enough in this case) can also produce electric field distortion under electric fields due to its significantly different dielectric constant from that of the polymeric matrix, thus achieving the same effect as other high-k fillers in improving ECE. In addition, there is no X-ray structural analysis of their nanocomposites by Chen et al., therefore the effects of the low-k (not as low as ours) fillers on the structural modification were still unclear at this point. We have now added a paragraph to discuss the advances of this work compared to Chen et al. which has been included in the reference list in the original submission. (Marked in red in page 5 of the revised manuscript.)

The reviewer is correct that the change in temperature-dependent dielectric spectroscopy is not severe. However, the phase transition we referred here is not the temperature-induced phase transition under the weak field, rather, it is the field-induced

structural phase transition at a fixed temperature. We have to emphasize that the ECE is the two entropy states that are reversibly connected by the external electric field. As we observed from the in-situ WAXD (**Fig. 2d-e**), the field-induced diffraction peak-shift was markedly enhanced for the ND-incorporated nanocomposite, compared to that for the base terpolymer. The peak broadening and shift indicate a small crystalline size (corroborated with the SAXS results in **Fig. 2c** and **Supplementary Fig. 14**), emerging multiphase-coexistence, and a strong field-induced structural modulation that would all contribute to the EC enhancement. Even under the weak field dielectric test, in addition to the frequency-dependent peaks that is normally indicating a relaxor ferroelectric state, we observed the emerging frequency-independent peaks at a higher temperature (~ 35 °C), indicating the slight formation of the polar phases that could be introduced by the organic-inorganic interface. Therefore, with all due respect, we believe the ND-incorporated nanocomposite facilitates the electric field-induced phase transitions and hence the ECE. To be clear, the title of the manuscript is changed to **“Low-k nano-dielectrics facilitate electric field-induced phase transition in high-k ferroelectric polymers for sustainable electrocaloric refrigeration”**.

Fig. R1. Temperature-dependent dielectric spectroscopy of the (a) base terpolymer and (b) ND-incorporated nanocomposite. The nanocomposite exhibited an emerging frequency-independent permittivity peak at ~ 35 °C, indicating an increasing polar structure in the polymeric matrix.

*MV·m⁻¹, $\Delta T \sim 19.8$ K, which is about 125% higher than that of the terpolymer ($\Delta T \sim 8.8$ K)), this work was not as excellent as the authors' claim (“These enhancements are even more effective compared to their high-*k* counterparts, i.e., the inclusion of 2.6 vol% of ND ... could increase the ECE for 60% at an electric field of 100 MV·m⁻¹ ... The value is nearly 70% higher than the best-reported high-*k* fillers⁴⁷”). Then, the ND nanocomposite had an obvious current leakage and conduction loss at high field, which was extremely unfavorable for the practical device applications.*

Response: We thank the reviewer for the comment. Compared to many high-*k*-filler nanocomposites, 60% of EC enhancement is achieved by incorporating merely 2.6 vol% of ND nanofiller. Here we would like to emphasize that an effective improvement of the ECE can be achieved with a small volume of nano-fillers. We compared the volumetric enhancement of ECE (the ratio between the EC enhancement in percentage and the volume percentage of the filler) at 100 MV·m⁻¹ and RT, for different nanofillers^{1, 3, 6}. As shown in **Fig. R2**, the ND introduced the highest volumetric EC enhancement (~23%/vol%) when compared with other fillers in the previous studies. We attributed this large volumetric enhancement to the relatively smaller size of the fillers of the NDs compared to many other 100-nm scaled ferroelectric ceramic particles. The comparison was shown in **Supplementary Section 2.4** and **Supplementary Fig. 8** in the original draft. We have added the clarification to the **Section Dielectric Analysis** to make it more accessible to the reader (marked in red in page 16 of the revised manuscript).

At this stage, we humbly recognized that the EC research has passed the stage of competing values of ECE. Among those extremely large EC values exhibited in high-*k*, ferroelectric filler incorporated nanocomposites in literature, we were interested in how to distinguish the possible contribution by the interface effect and by the ECE of the extrinsic fillers. In this work, ND was chosen for its EC-inert nature and low dielectric constant compared to the polymeric matrix. This EC enhancement of the nanocomposite may be dominated by the abrupt permittivity change-induced interface effect, which generates a local distorted field and reduced the energy barrier of the field-

induced phase transition. Therefore, both high-k and low-k fillers (relative to the polymeric matrix) should work in a similar way to enhance ECE.

Fig. R2. Volumetric EC enhancement of different fillers.

We strongly agree with the reviewer that the current leakage and conduction loss at the high electric field are unfavorable for practical device applications. We directly tested the heat flux across the EC nanocomposite film with a heat flux sensor (**Fig. R3**).

Fig. R3. Heat flow intensity of the base terpolymer and T-ND-2.6% at $100 \text{ MV}\cdot\text{m}^{-1}$.

The results indicate that the T-ND-2.6% exhibited no obvious joule heating during the whole time of application of the electric field ($100 \text{ MV}\cdot\text{m}^{-1}$). In addition, the difference between the heat ejection and absorption in one EC cycle is not severe. We acknowledge that the inclusion of the extrinsic nanofillers would in general cause higher conduction loss, fortunately, the directly measured EC results indicate the situation is mild and wouldn't affect the device operation.

Above discussion and results have been added in the revised **Supplementary Section 2.4**.

3. In Fig. 2f, when the electric field was zero, the proportion of the polar phase to 0 % was not rigorous. Undetectable by XRD did not mean that the phase was not present.

Response: The reviewer is correct. We were not intended to identify the exact 0% of polar phases under zero-field. Therefore, in **Fig. 2(f)** of the revised manuscript, we change the vertical coordinate to “**Field-induced polar phase**”. (See **Fig. R4**)

Fig. R4. Fraction of field-induced polar phase as a function of the electric field.

4. The particle size of nano-diamond affected the interface area and polarization. From the results of AFM and SEM, the diameter distribution of the ND was about 50-100 nm, which was not consistent with the 10 nm mentioned by the authors.

Response: Thanks for the kind suggestions. The nano-diamond products we purchased from **3A company** were labeled to exhibit particle sizes in the 5~10 nm range, which has been verified by the TEM results (**Fig. 1b**). When the NDs are dispersed into the polymer, some of the NDs would inevitably agglomerate together. The TEM results for T-ND-2.6% are shown in **Fig. R5**.

Therefore, in the nanocomposite films, there are individually dispersed NDs and also particles agglomerated together (the size is 20~100 nm). The individually dispersed

NDs are easier to be observed by the TEM rather than that by the SEM and the AFM we utilized. We have added the TEM images of NDs inside the polymeric matrix in **Section 2.11** of the revised **Supplementary Material** (see **Supplementary Fig. 21**).

Fig. R5. TEM images of T-ND-2.6%. **(a-b)** individually distributed and **(c)** agglomerated NDs observed inside the polymeric matrix.

5. Compared with the terpolymer matrix, the melting enthalpy of the nanocomposite film decreased, but the crystallinity in Supplementary Table 1 increased. And as the electric field increased, the grain size of the polar phase increased first and then decreased. Was there a problem with the fitting method used?

Response: We thank the careful observation from the reviewer. Initially, we calculate the melting enthalpy using the mass of the EC composites. Only the polymer undergoes a phase change during the melting process, so the mass of the polymer should be used in the calculation, and the results are shown in **Fig. R6**. (Corrected in **Supplementary Fig. 12a**) It is noteworthy that the crystallinity and melting enthalpy of the EC nanocomposites were slightly enhanced compared to the base terpolymer.

We thank the reviewer for the in-depth comments on the grain size of the field-induced polar phases. The previous calculation of the polar phase grain size was incorrect owing to an inexcusable miscalculation. Fortunately, the error does not affect the macroscopic conclusion that the ND-incorporated nanocomposite exhibited stronger phase transition under the same field compared to the base terpolymer. The grain size of the polar phase gradually increases as the electric field increases, as shown in **Table R1**. We have corrected the grain size of the polar phase in **Supplementary**

Table 1. Since the polar phase cannot be accurately captured in the zero-field state, the grain size of the polar phase at this time cannot be obtained.

Fig. R6. DSC profiles of the samples recorded during the heating scan.

Table R2. Crystallinity, fraction of non-polar phases and polar phases, and crystal sizes of terpolymer and T-ND-2.6%

Sample	Terpolymer- 0 MVm ⁻¹	Terpolymer- 50 MVm ⁻¹	Terpolymer- 80 MVm ⁻¹	T-ND-2.6%- 0 MVm ⁻¹	T-ND-2.6%- 50 MVm ⁻¹	T-ND-2.6%- 80 MVm ⁻¹
Crystallinity	33%	41%	45%	36%	43%	48%
Field- induced polar phases	0%	8%	18%	0%	16%	31%
Crystal size of NP phases	35.4 nm	34.1 nm	32.4 nm	29.0 nm	26.5 nm	21.1 nm
Crystal size of polar- phases		8.8 nm	9.1 nm		12.2 nm	13.1 nm

6. In page 9, the authors proposed that “The thermal conductivity is attributed to the large thermal conductivity of the ND particles to reduce the thermal resistance within the polymeric matrix.” However, the effect was not significant. Why were the excellent characteristics of high thermal conductivity fillers not reflected? In addition to the percolation theory, what were the possible reasons? If increasing the proportion of the nano-diamond composite could improve the thermal conductivity, what effect would it have on the electrocaloric performance of the system?

Response: We thank the reviewer for the in-depth comments. The main objective of our study is to investigate the effect of low-k nano-dielectrics on the electric-induced phase transition of the EC polymers. In this study, 2.6 vol% of NDs were doped to improve the electrocaloric cooling performance, thermal conductivity, and dielectric insulation properties of the EC polymers.

The thermal conductivity is improved by 75% in the study, which is not too bad considering the small volume concentration of the ND. The reason for the lack of significant improvement in thermal conductivity is that the amount of NDs doped is too small to form an interconnected heat transfer network. Further increase in NDs fillings will lead to a sharp decrease in the ECE of the ND-incorporated nanocomposites due to more pronounced agglomeration, as shown in **Fig. R7**.

Fig. R7. The ECE of the base terpolymer and ND-incorporated nanocomposites. **(a)** EC-induced ΔS . **(b)** EC-induced ΔT .

7. In page 10, the authors cited the literature on boron nitride for comparison (Literature 54), and the cited article is incorrect. The authors stated that “Different from boron nitride nanosheet (BNNS), the other filler with high thermal conductivity, low- k and high dielectric strength that has been widely applied in EC polymer as fillers, the ND-incorporation observably enhanced the ECE of the base terpolymer rather than deteriorate the ECE as the BNNS does⁵⁴”. Literature 54 does not study the impact of the BNNS on the ECE, much less the deterioration described by the authors.

Response: We thank the reviewer for the sharp observation. BNNS is known for its high thermal conductivity owing to the highly ordered 2-D atomic structure. As a result, the BNNS-incorporated P(VDF-TrFE-CFE) was expected to show smaller permittivity and reduced polarization. Literature 54⁷ (Li Q, Zhang G, Liu F, et al. Energy & Environmental Science, 2015, 8(3):922-931.) demonstrated the reduction in the polarization of the composites formed by the base polymer with BNNS (**Fig. 4(a)** in the literature, shown in **Fig. R8**). These reductions of dielectric responses are not due to the reduction of the polar correlation in a high polar-entropy polymer⁸, which is in favor of EC enhancement but is more likely due to the BNNS-induced ordering near the polymer chain-BNNS interface under the zero-field, which could dramatically reduce the field-induced entropy change.

Fig. R8. The electric displacement of the base terpolymer and BNNS-incorporated nanocomposites in the literature⁷.

The reduced polarization in **Fig. R8** was a hint of the EC reduction that could also be predicted by the Landau phenomenological theory. ($\Delta S = -\frac{1}{2}\beta(P^2(E_h) - P^2(E_l))$)

In addition, Zhang et al. demonstrated¹ that the Q , EC-induced ΔT and EC-induced ΔS decrease as the content of BNNSs increases when they tested the EC performance of the P(VDF-TrFE-CFE)/BNNSs nanocomposites. (**Fig. S22** in the literature¹, shown in **Fig. R9**)

Fig. R9. The ECE of P(VDF-TrFE-CFE) and P(VDF-TrFE-CFE)/BNNSs nanocomposites in the literature¹.

To verify the above hypothesis, we also prepared composites by blending polymers with BNNS and tested the electrocaloric cooling performance using the direct method, as shown in **Fig. R10**. The ECE in the nanocomposite was markedly reduced as predicted. Therefore, we believe that blending the polymer with BNNS will deteriorate its electrocaloric cooling performance.

We now changed this sentence to “Different from boron nitride nanosheet (BNNS), the other filler with high thermal conductivity, low-k, and high dielectric strength that has been widely applied in EC polymer as fillers, the ND-incorporation observably enhanced the polarization of the base terpolymer rather than deteriorate the polarization as the BNNS does. (This is likely to indicate that the incorporation of BNNS reduces the ECE of the terpolymer)”. (Marked in red on page 9 of the revised manuscript.)

Fig. R10. Comparison of the ECE of the base terpolymer and BNNS-incorporated nanocomposites.

We have added the above discussion and **Fig. R10** (see **Supplementary Fig. 22**) to **Supplementary Section 2.12**.

8. In page 15, the authors mentioned that “To single out the orientation polarization, we subtracted the space charge-induced polarization to correctly reveal the polarization-entropy correlation”. How to accurately calculate the space charge-induced polarization?

Response: Thanks for the reviewer’s kind suggestions. What we intended to say here is that we removed the conduction loss from the measured P-E loop, but we referred to it as the space-charge-induced polarization by mistake. The conduction loss from the leakage current contributes to the nominal polarization that was measured, so we subtracted the contribution from the measured nominal polarization to obtain the real polarization. We have now modified this sentence to “To single out the polarization that contributes to the ECE, we identified and removed the contribution from conduction loss to obtain the polarization-entropy correlation” (marked in red on page 15 of the revised manuscript). The identification of the conduction loss and the method for removing it is as follows.

The polymeric film is not an ideal insulator and leakage current is inevitable which induces conduction loss. As shown in **Fig. R11**, in unipolar P-E loops, when the applied

electric field is cycled once and returned to zero, the measured nominal polarization returns a finite nonzero value ΔP . For the PVDF-based terpolymer ferroelectric relaxor studied here, the contribution of polarization relaxation to the “remnant polarization” after an electric field cycle should be negligible⁹, indicating that the observed nonzero ΔP is dominated by the conduction loss.

Fig. R11. Unipolar P-E loops of the base terpolymer before and after conduction subtraction⁹.

Furthermore, in our unipolar P-E loop measurement, through a test of decreasing the applied field frequency, we find a monotonous increase of $\Delta P(E=0)$, as shown in **Fig. R12**. Such an increase of $\Delta P(E=0)$ is consistent with the accumulation of charge leakage upon increasing the time duration, further confirming the nature of the conduction loss.

Fig. R12. Unipolar P-E loops of T-ND-2.6% under different frequencies.

The conductivity σ of the terpolymer may be calculated from the unipolar P-E loops through:

$$\Delta P = \frac{2 \int_0^{E_{\max}} \sigma E dE}{E_t} \quad (\text{Eq. R1})$$

where E_t is the ramp rate of the applied electric field (following a unipolar triangular waveform), E is the applied electric field and E_{\max} is the maximum applied electric field.

$$E = E_t * t \quad (\text{For the charging cycle when the field is increased}) \quad (\text{Eq. R2})$$

$$E = E_{\max} - E_t (t - T / 2) \quad (\text{For the discharging cycle when the field is decreased}) \quad (\text{Eq. R3})$$

$$E_{\max} = E_t * T / 2 \quad (\text{Eq. R4})$$

where t is the time at which the electric field E is applied and T is the period of the unipolar triangular applied electric field.

From Eq. R1, the conductivity σ can be calculated. Then the contribution of conduction to the polarization P_c under a certain electric field E_a on the P-E loops can be calculated as:

$$P_c = \int_0^{E_a} \sigma E dE / E_t \quad (\text{For the charging cycle when the field is increased}) \quad (\text{Eq. R5})$$

$$P_c = \int_0^{E_{\max}} \sigma E dE / E_t + \int_{E_a}^{E_{\max}} \sigma E dE / E_t \quad (\text{For the discharging cycle when the field is decreased}) \quad (\text{Eq. R6})$$

Based on the above equations, the contribution from conduction to the P-E loops can be subtracted from the curves. The corrected polarization returns to zero after the electric-field cycle, as shown in **Fig. R11**. It is then possible to study the response of the polarization and its contribution to the electrocaloric cooling performance. The detailed correction method was added to **Supplementary Section 2.8**.

9. *In the Landau–Devonshire phenomenological theory (equation. (1)), was it applicable to both bulk and composite materials? If applicable, how to clearly distinguish dipole polarization from space charge-induced polarization in this formula? Did the electrocaloric effect necessarily come from the dipole steering polarization and*

the interface effect?

Response: The Landau–Devonshire phenomenological theory given in Equations (1)-(2) is applicable to both bulk and composite materials. For composite materials, quantities like β , Θ , and Ω in Equations (1)-(2) should adopt the effective values of those of the composite.

After subtracting the contribution of conduction loss (which we originally referred to as space-charge-induced polarization) from the measured nominal polarization, we believe that the corrected polarization is dominated by the dipole polarization, with very small contribution from space charges. This is shown by performing a frequency test of the polarization and electrocaloric response described below.

As shown in **Fig. R13**, we find that unipolar P-E loops show very close peak polarizations across different frequencies of 10, 100, and 1000 Hz after deducting the conduction loss. Since space charge is only responsive at low frequencies and its contribution is eliminated in high-frequency measurements, the result indicates a very small contribution of space charge to the polarization response.

Fig. R13. Unipolar P-E loops (a) and corrected unipolar P-E loops (b) of T-ND-2.6% under different frequencies.

In addition, as shown in **Fig. R14**, we find that the ramping time of applying and removing the electric fields, across 0.1, 0.01, and 0.001 s, show a very minor effect on the cooling performance of the EC materials (i.e., less than 5% difference in heat flux signal). Due to the slow response of space charges, this ramping-time test result of ECE suggests that the ECE is almost entirely contributed by the dipole polarization.

Fig. R14. EC heat flux signals under different ramping times for applying and removing the field of (a) 0.1 s, (b) 0.01 s, and (c) 0.001 s.

Therefore, we conclude that the ECE is dominated by the dipole contribution, and the measured polarization and ECE presented in our current work correctly convey their correlation. The dipole polarization is further modified by the interfacial effect³, thus giving rise to an enhancement of the ECE.

10. *The authors demonstrated the advantages of the composite materials only through the simulation of the fluid-solid coupling-based EC refrigerator, lacking experimental results of real devices. Then, both the device structure and the heat transfer simulations in SI have been published in the authors' earlier work (Applied Thermal Engineering 2021, 190, 116806).*

Response: The purpose of our numerical simulation is to use the simulation platform to demonstrate the advantages of ND-incorporated nanocomposites over the base terpolymer (ECE and heat transfer performance). Any EC materials can be evaluated for comprehensive performance using the simulation platform. Our earlier work¹⁰ (Applied Thermal Engineering 2021, 190, 116806) was mainly intended to show the effect of system layout and fluid flow direction under different operating strategies on the cooling performance of the EC device.

Thanks for the reviewer's kind suggestions. We have recently added some experimental studies to further evaluate the comprehensive performance of the base terpolymer and EC nanocomposites. We developed a film-like EC oscillating refrigerator that is in a similar configuration and operating strategy as the previously reported EC device¹¹. The EC device was utilized as a standard platform to evaluate the

material properties and their relation to the device's performance. **Figures R15(a)** and **R15(b)** respectively give the side views of the device. The EC film was sandwiched and moved between two surface-insulated stainless-steel sheets driven by electrostatic forces that alternatively changed with the change of the electric field applied to the EC film. Two ceramic plates kept at constant temperatures were placed behind the sheets serving as hot and cold ends, with two heat flux sensors (firstly calibrated) between them to measure the heat flux signals.

Fig. R15. The side views of our EC device under (a) Zero field and (b) Applied field.

Figure R16 gives the detailed calibrating and measuring processes for the heat flux signal measurement. The sensors were calibrated using a heating plate with a constant resistance (R) before the measurement as shown in **Fig. R16**. The resistance was placed on the ceramic plate, with the sensor sandwiched between them, and foam was placed on the resistance for thermal insulation. Initially, the sensor gave a constant signal U_0 (**Fig. R16**). As a voltage (U) was applied to the resistance, Joule heat ($P_{\text{Joule}}=U^2/R$) was transferred from the resistance to the ceramic plate through the sensor, causing a changed signal U_1 . We got the change value in voltage ($U_{\text{change}}=U_1-U_0$) of the sensor proportional to the Joule heat as the signal was stable, and the proportional factor γ between heating power density and sensor signal could be calculated as

$\gamma = PJ_{\text{oule}} / (S * U_{\text{change}})$ in which S is the area of the resistance.

Fig. R16. The calibration process for heat flux sensors in the EC device.

In the device, as the EC film alternatively attached to cold and hot ends shown in **Fig. R15**, signals appeared as U_{cooling} , and the original signals are provided in **Fig. R17**. The corresponding heating (HPD) and cooling (CPD) power densities could be calculated as $HPD = \gamma * U_{\text{heating}} / (t_h * \rho)$ and $CPD = \gamma * U_{\text{cooling}} / (t_h * \rho)$ in which t_h is the thickness of the EC film (30 μm) and ρ is the density.

Fig. R17. The heat flux signal measurement in the EC device.

The experimental results of this platform are intended to illustrate the superior cooling capacity (the same CPD can be obtained at lower electric fields) and the better

heat transfer performance (shorter heat dissipation time) of the ND-incorporated nanocomposites compared to the base terpolymer. We have added the experimental details of such an EC device to **Supplementary Section 4**.

11. *The notes in the data chart were inconsistent with the text, please check it carefully (Fig. 5 T-TD-2.5%, T-TD-5%?). The annotation information of the sample loss value in Supplementary Fig.13d was wrong.*

Response: We thank the reviewer for the careful examination and kind notice. We have now corrected the notes in **Fig 5** (T-ND-2.5%, T-TD-5%, by weight) to T-ND-1.2% and T-ND-2.6% (by volume). In addition, we have also double-checked the revised manuscript to ensure accuracy.

Reviewer #2 (Remarks to the Author):

Authors reported the well-studied P(VDF-TrFE-CFE), and introduced a low content of nanodiamond (low dielectric constant) to improve the overall performance of the EC polymers. It's interesting that nanodiamond can promote the interface polarization and reduce the phase transition energy barrier due to the significantly different dielectric constant from the polymeric matrix. The entropy change of T-ND-2.6% is improved by 60% as compared to the already existed high entropy change in the base terpolymer. In addition, the thermal conductivity and dielectric insulation of the ND-incorporated nanocomposites have also been improved to varying degrees. Overall, the paper is well written with some interesting findings, and the comparison based on the current results and reported literatures are performed. However, some comments/questions need to be amended prior to the final acceptance. Detailed comments are included as below.

(1) The direct method is a widely used technique for quantifying the EC-induced temperature changes. As mentioned in the manuscript that the temperature change of the EC materials is obtained by using the infrared camera, showing the slightly smaller value than that of obtained by the direct method. Can authors please provide the explanation? Moreover, how to ensure the accuracy of the infrared camera test?

Response: We thank the reviewer so much for acknowledging our work. To validate the specific EC measurement setup built in our facility at Shanghai Jiao Tong University, we further calibrated the ECE measurement instrument with an infrared camera. In the direct EC temperature-change measurement (**Supplementary Fig. 9**), the ramping time of applying and removing the electric fields was kept at 0.1 s (rather than suddenly applying and removing the fields). Therefore, the measurement condition is not a strict adiabatic condition⁸. As a result, the recorded temperature change should not represent 100% of the adiabatic temperature change, but less (**Supplementary Fig. 10** and **Supplementary Fig. 11**).

A Fourier Transform Infrared (FTIR) spectrometer with a gold integrating sphere

(Nicolet 6700, THERMO FISHER) was employed to get the accurate value of emissivity of the polymeric film (see **Fig. R18**), which ensures the accuracy of our infrared camera results.

Fig. R18. Schematic diagram of the IR camera for temperature changes of EC materials and the corresponding emissivity at mid-infrared wavelengths (7-13 μm).

(2) *Adding some defects in the base terpolymer helps the crystallinity, which is also observed in this work and this crystallinity of course improves the EC performances. My question is what is the specific contribution of crystallinity when compared to the contribution of polar/non-polar transition? Further analysis is required.*

Response: We thank the reviewer for the in-depth comments. T-ND-2.6% shows slightly higher crystallinity (less than 10% increase in crystallinity) compared with the base terpolymer. The increase of ΔS at $100 \text{ MV} \cdot \text{m}^{-1}$ of T-ND-2.6% is 60% compared with the base terpolymer, which is much larger than the change of crystallinity. Hence, it is the contribution of polar/non-polar that generates a large improvement in the EC cooling performance.

In this work, by introducing low-k nano-dielectrics (nanodiamond, ND), we

discovered a unique way to slightly enhance the crystallinity while lowering the phase change energy barrier. The easier field-induced phase transition from a high-polar-entropy state induced better ECE, as shown in **Table R2**.

Table R2. Crystallinity, fraction of non-polar phases and polar phases, and crystal sizes of terpolymer and T-ND-2.6%

Sample	Terpolymer- 0 MVm ⁻¹	Terpolymer- 50 MVm ⁻¹	Terpolymer- 80 MVm ⁻¹	T-ND-2.6%- 0 MVm ⁻¹	T-ND-2.6%- 50 MVm ⁻¹	T-ND-2.6%- 80 MVm ⁻¹
Crystallinity	33%	41%	45%	36%	43%	48%
Field- induced polar phases	0%	8%	18%	0%	16%	31%
Crystal size of NP phases	35.4 nm	34.1 nm	32.4 nm	29.0 nm	26.5 nm	21.1 nm
Crystal size of polar- phases		8.8 nm	9.1 nm		12.2 nm	13.1 nm

(3) Compared to other refrigeration technologies, the TEWI values of EC materials are significantly lower. For instance, the improved ECE, EC strength, and temperature independency in the T-ND-2.6% could lower the overall CO₂ emission of a potential EC device. It would be useful for the authors to give how the TEWImat is calculated so that this reported results can be more easily understood for a wider audience.

Response: Thanks for the kind suggestions. The total equivalent warming impact

(TEWI) can be used to evaluate any technology that consumes energy, as long as the energy has an origin associated with the CO₂ emission. In conventional VCR, because of the leakage of the high GWP of the refrigerant, TEWI is commonly composed of two parts: (1) the direct emission of strong GWP vapors and (2) the electricity power it consumes, which can be calculated to the equivalent CO₂ emission as an indirect contribution to the TEWI.

$$\text{TEWI} = \text{CO}_{2, \text{direct}} + \text{CO}_{2, \text{indirect}} \quad (\text{Eq. R7})$$

where CO_{2, direct}, CO_{2, indirect} are the direct and indirect equivalent CO₂ emission.

$$\text{CO}_{2, \text{direct}} = x * \text{GWP} \quad (\text{Eq. R8})$$

$$\text{CO}_{2, \text{indirect}} = \frac{Q}{\text{COP}} * T_{\text{CO}_2} * L \quad (\text{Eq. R9})$$

where x is the quantity (in weight) of the vapor emission, Q the cooling capacity of the refrigeration system, T_{CO_2} the average mass of carbon dioxide released per kWh of electrical energy in power plants, and L the lifetime (in years) of the refrigeration system, respectively.

The above assumptions allow the TEWI (as a device parameter) to degenerate to a material performance index TEWI_{mat}. The indirect TEWI_{mat} is directly related to the material COP (COP_{mat}) of the EC material, as previously summarized by Qian et al. in 2016¹². For caloric cooling, COP_{mat} can be depicted as:

$$\text{COP}_{\text{mat}} = \frac{Q}{W_{\text{net}}} = \frac{T_C \Delta S - \frac{1}{2} Hy}{(T_H - T_C) \Delta S + Hy} \quad (\text{Eq. R10})$$

where W_{net} is the net input work per unit mass, T_C/T_H the temperature at the cold/hot end, Hy the hysteresis in caloric materials.

We have added the definition and calculation of TEWI to the revised manuscript (Listed after **Fig. 4**, marked in red in page 17-18 of the revised manuscript). In addition, we have carefully checked the manuscript and explained all the variables for better understanding.

(4) *In this work, numerical simulations were carried out on the designed rotary EC device. The authors claim to have performed a grid-independent verification in the Supplementary Section 3.2. It is recommended to provide the detailed results and discussion of the grid independence verification to guarantee the accuracy of the simulation.*

Response: We thank the reviewer for the kind suggestions. In this study, mesh-independence verification and the selection of a suitable mesh can help speed up the computational process. Models with grid numbers 282640, 1573000, 3714000, and 6842000 were constructed, respectively.

As shown in **Fig. R19**, the grid-independent validation results show that when the number of grids exceeds 3714000, the average temperature of the cold fluid outlet (T_{cold}) hardly changes anymore (less than 0.1 K). Therefore, the model with the grid number 3714000 was chosen for the calculation.

Fig. R19. Grid-independent verification.

We have added the results of the grid-independence verification in **Supplementary Section 3.2** (see **Supplementary Fig. 24**). (Marked in red)

Reviewer #3 (Remarks to the Author):

The authors report on the improved electrocaloric properties of the ferroelectric terpolymer-based nanocomposites with nanodiamond fillers. These fillers with low dielectric constant not only somehow cause interfacial polarization and boost electrocaloric effect but also help to increase the thermal conductivity of the terpolymer to result in better cooling performance in cooling devices.

While the study presents some novel, interesting and counterintuitive results, several points need to be clarified before a decision can be made. Most importantly, the language of the manuscript is not clear therefore important messages cannot be conveyed enough. Significant improvement of the discussion and the writing is necessary.

1)The manuscript has some significant overlap with ref 11. The authors are suggested address this more and stress the differences and similarities more clearly. They state that the number of polar entities do not change or even decrease in the current manuscript unlike what was reported in ref. 11. Instead, they attribute the increased ECE to the increased polarization induced at the interfaces. However, similar to ref.11, here, the authors also suggest that reduced energy barriers for field-induced non-polar to polar transition. Is this not playing role in the increased ECE? In another section, this is also held responsible. In other words: The origin of the enhanced electrocaloric effect is unclear from the discussion. Is it the field-induced non-polar polar phase transition, which is facilitated by the ND fillers, or is it the interface effect? The current discussion is confusing.

Also, the title of the manuscript can be more exact reflecting this facilitated field-induced transition: What the nanodiamond filler facilitates or modulates is the electric-field induced phase transition.

Response: We thank the reviewer for the in-depth comments. In the study, the combined effect of the reduced phase transition energy barrier of the EC polymer and

the interfacial effect (larger polarization) leads to the improved electrocaloric performance of the ND-incorporated nanocomposites compared to the base polymer, as shown in **Fig. R20**. Unlike the ND-incorporated nanocomposites in the study, the earlier work⁸ (ref. 11, **Qian X, et al. Nature 600, (2021).**) produced materials with slightly lower polarization compared to the base polymer, but with much lower phase transition energy barriers (see **Fig. R21**), which also resulted in materials that exhibit significantly better EC performance.

Fig. R20. Comparison of polarization and field-induced phases of ND-incorporated nanocomposite and the base terpolymer in this study.

Fig. R21. Comparison of polarization and polar phase transition ratios of TD-0.6% and the base terpolymer in our earlier work (ref11)⁸.

Thanks for the reviewer's kind suggestions. We have changed the title of the manuscript to "**Low-k nano-dielectrics facilitate electric-field induced phase transition in high-k ferroelectric polymers for sustainable electrocaloric refrigeration**"

2) In Fig. 2a the authors show WAXD spectra of the terpolymer & nanocomposites and state that after ND incorporation, 'the peaks for the non-polar phase of the nanocomposites shifted to a lower angle.' They do not specify the non-polar or polar phases in general in the manuscript. Is the non-polar one α , polar one β ? More information is necessary about this section.

Response: Thanks for the kind suggestions. The crystal types of the base terpolymer and PVDF are similar, but not identical. For example, the non-polar phase in P(VDF-TrFE-CFE) is not a complete TGTG structure but can be considered non-polar. Therefore, it can be considered that the non-polar phase is close to the α -phase and the polar phase is close to the β -phase, as shown in **Fig. R22**.

As suggested, we have added the explanation about the non-polar phase and polar phase to the revised manuscript and **Supplementary Section 2.7** to make it more accessible to the reader. (Marked in red)

Fig. R22. Estimates of crystallinity and fractions of non-polar, and polar phases.

In addition, we have carefully checked the manuscript and explained all the variables (β coefficient, TEWI et al) in the manuscript for better understanding.

3) It is indeed surprising that low- k filler results in a similar interface effect as the high- k filler. The interface that forms between high- k filler and the polymer typically amplifies the external electric field due to the contribution from the easily polarizable

high-k filler. How do the authors explain the observation of a similar interface effect with the low-k filler? How significant is the built-up potential at the interface shown in Fig.3a-b. Can the authors compare this with the same potential reported at the interface between the polymer and high-k filler?

Response: We thank the reviewer so much for acknowledging the interesting findings in this work. The field distortion has been discovered in many EC nanocomposites and was believed to contribute to EC enhancement. Our earlier reports verified the theory by developing a phase-field model to study the EC nanocomposites¹³. From Maxwell equations, the electric field will be distorted at any interface as long as two sides have different permittivity (See **Fig. R23**). Therefore, our simple hypothesis is that the ECE will be enhanced at the interfaces in the polymer nanocomposites with both high-k and low-k fillers (“higher” or “lower” than the permittivity of the polymeric matrix), if there is no other chemical or physical interaction at the interface playing a key role. It is a surprise to us that most of the studies solely focused on the high-k fillers (*i.e.*, mostly ferroelectric ceramics), rather than the low-k fillers, which should have a similar effect of field distortion.

Fig. R23. Schematic electric field distortion around a spherical filler with a permittivity higher (a) or lower (b) than the polymeric matrix. The applied electric field is $100 \text{ MV} \cdot \text{m}^{-1}$.

On the other hand, we know that most of the polar ceramics exhibit excellent electrocaloric effects. Incorporating these ceramic nanoparticles in the polymer would cause complications in identifying the origin of the possible EC enhancement, *i.e.*, it

will be unclear whether the mechanism of the enhancement of the ECE in composites is due to the ECE of the nanofillers, or the electric field distortion, or both.^{1, 14, 15}

In this study, by using the EC-inert nano-diamonds as fillers, we developed the nanocomposite to solely study the interfacial field distortion and its effect on the ECE. We proved that low-k filler could effectively induce field distortion that significantly enhances ECE. As expected, we found that the smaller fillers (10 nm scales) were more effective in enhancing the ECE than the larger ones (100 nm scales), by introducing more interface area in the composite, which further corroborated the effectiveness of the field-distortion at the interface.

Due to the different AFM tools applied for mapping the interface potential, as well as different material systems (polymer and fillers), it would be difficult to quantitatively compare **Fig. 3b** and the ones in the literature^{3, 16}. At this stage, we have no evidence to identify which filler is more effective, however, we believe smaller-sized fillers could induce more interfacial area and hence enhance ECE more effectively as we showed in **Fig. R2** (copied below for the convenience of the reviewer, also see in **Supplementary Fig. 8**).

Fig. R2. Volumetric EC enhancement of different fillers.

4) *It is quite difficult to conclude from the SEM image in Fig. 1c that the dispersion of the nanoparticles in the polymer matrix is homogeneous. The nanoparticles are very small but still...*

Response: Thanks for the kind suggestions. The nano-diamond products we purchased from **3A company** were labeled to exhibit particle sizes in the 5~10 nm range, which has been verified by the TEM results (**Fig. 1b**). When the NDs are dispersed into the polymer, some of the NDs would inevitably agglomerate together. The TEM results for T-ND-2.6% are shown in **Fig. R24**.

Therefore, in the nanocomposite films, there are individually dispersed NDs and also particles agglomerated together (the size is 20~100 nm). The individually dispersed NDs are easier to be observed by the TEM rather than that by the SEM and the AFM we utilized. We have added the TEM images of NDs inside the polymeric matrix in **Section 2.11** of the revised **Supplementary Material** (see **Supplementary Fig. 21**).

Fig. R24. TEM images of T-ND-2.6%. **(a-b)** individually distributed and **(c)** agglomerated NDs observed inside the polymeric matrix.

5) On line 157, it is stated that ‘...electrical breakdown field of EC nanocomposites, as evidenced by Fig. 1d’. Figure number is confused apparently.

Response: Thanks for the kind reminder. This was a clerical error and the results of dielectric breakdown are shown in **Fig. 1e**. We have also double-checked the manuscript and Supplementary materials to ensure the accuracy.

6) The authors do not describe what is shown in Fig. 2c adequately. How can the reader agree with their conclusion that long-range ordering is suppressed if they don't describe what they observe in the Figure for those who are not familiar with SAXS? This is a general problem. The way the authors write the manuscript does not help the reader enough to understand what the figures tell or how did the authors draw

conclusions from the data in the figures.

Response: Thanks for the reviewer's kind suggestions. A detailed description has been added in the main text (marked in red in page 12 of the revised manuscript). In **Fig. 2c** of the original draft, we showed the raw data from the SAXS test. In detail, we used the Lorentz-corrected SAXS spectra in the calculation of the long period. Evidence for crystal thinning in the chain direction is obtained from the small-angle X-ray scattering (SAXS) data presented in **Fig. R25**, showing a correlation peak characterizing the period organization of the crystalline lamellae. The long period, L_p , provides information on the periodic arrangement of the crystalline lamellae and amorphous regions. All the details of the SAXS analysis used in this study, for semi-crystalline polymers were already published^{17, 18}.

$$L_p = \frac{2\pi}{q} \quad (\text{Eq. R11})$$

Fig. R25. Lorentz-corrected SAXS spectra, $q^2 \cdot I(q)$, measured for the base terpolymer and ND-incorporated nanocomposites.

The reduction of the peak signal corresponding to the long cycle indicates that NDs may suppress the long-range ordering in the polymeric matrix. We have added the processing about SAXS to **Supplementary Section 2.7**.

In addition, we have added captions to all figures in the manuscript that are not explained in detail to make them more accessible to the readers. (**Fig. 1b, c, Fig. 2b, c, Supplementary Section 2.7, Fig. 3c, d, Supplementary Section 2.8, Fig. 4c and Supplementary Section 2.9**, marked in red)

7)Another example to the above problem: What is the β coefficient, what does it represent? Not clearly written.

Response: Thanks for the reviewer's kind suggestions. Here, β coefficient is the temperature-independent Landau–Devonshire coefficient, which can be obtained by fitting to the dielectric properties. Specifically, β is the temperature derivative of the reciprocal permittivity¹⁹. We have added the explanation to the manuscript (In **Section dielectric analysis**, marked in red).

$$\beta = \frac{\ln\Omega}{\varepsilon_0\Theta} \quad (\text{Eq. R12})$$

where Θ is an effective Curie constant, and Ω is the number of accessible polar orientations, and ε_0 is the vacuum permittivity while ε_r is the permittivity of EC materials.

In addition, we have carefully checked the manuscript and explained all the variables in the manuscript for better understanding.

Reference

1. Guangzu, *et al.* Ferroelectric polymer nanocomposites for room-temperature electrocaloric refrigeration. *Advanced materials (Deerfield Beach, Fla)*, (2015).
2. Zhang G, *et al.* Ferroelectric polymer nanocomposites with complementary nanostructured fillers for electrocaloric cooling with high power density and great efficiency. *ACS Applied Energy Materials* **1**, 1344–1354 (2018).
3. Qian J, *et al.* Interfacial Coupling Boosts Giant Electrocaloric Effects in Relaxor Polymer Nanocomposites: In Situ Characterization and Phase-Field Simulation. *Adv Mater* **31**, e1801949 (2019).
4. Kang X, Jia S, Peng J, Yu H, Zhou X. Electromagnetic-driven electrocaloric cooling device based on ternary ferroelectric composites. *Composites Part B: Engineering* **227**, 109391 (2021).
5. Chen X-Z, *et al.* A nanocomposite approach to tailor electrocaloric effect in ferroelectric polymer. *Polymer* **54**, 5299–5302 (2013).
6. Chen Y, *et al.* An All-Scale Hierarchical Architecture Induces Colossal Room-Temperature Electrocaloric Effect at Ultralow Electric Field in Polymer Nanocomposites. *Adv Mater* **32**, e1907927 (2020).
7. Li Q, *et al.* Solution-processed ferroelectric terpolymer nanocomposites with high breakdown strength and energy density utilizing boron nitride nanosheets. *Energy & Environmental Science* **8**, 922–931 (2015).
8. Qian X, *et al.* High-entropy polymer produces a giant electrocaloric effect at low fields. *Nature* **600**, (2021).
9. Chu B. *PVDF-based copolymers, terpolymers and their multi-component material systems for capacitor applications*. The Pennsylvania State University (2008).
10. Li Q, Shi J, Han D, Du F, Chen J, Qian X. Concept design and numerical evaluation of a highly efficient rotary electrocaloric refrigeration device. *Applied Thermal Engineering* **190**, (2021).
11. Ma R, *et al.* Highly efficient electrocaloric cooling with electrostatic actuation. *Science* **357**, 1130–1134 (2017).
12. Qian S, *et al.* Not-in-kind cooling technologies: A quantitative comparison of refrigerants and system performance. *International Journal of Refrigeration* **62**, 177–192 (2016).

13. Yang L, *et al.* Graphene enabled percolative nanocomposites with large electrocaloric efficient under low electric fields over a broad temperature range. *Nano Energy* **22**, 461-467 (2016).
14. Zhang G, *et al.* Ferroelectric Polymer Nanocomposites with Complementary Nanostructured Fillers for Electrocaloric Cooling with High Power Density and Great Efficiency. (2018).
15. Lu Y-C, *et al.* Enhanced electrocaloric effect for refrigeration in lead-free polymer composite films with an optimal filler loading. *Applied Physics Letters* **114**, (2019).
16. Simin, *et al.* Direct Detection of Local Electric Polarization in the Interfacial Region in Ferroelectric Polymer Nanocomposites. *Advanced Materials* **31**, 1807722-1807722 (2018).
17. Le Goupil F, *et al.* Enhanced Electrocaloric Response of Vinylidene Fluoride - Based Polymers via One - Step Molecular Engineering. *Advanced Functional Materials* **31**, 2007043 (2021).
18. Tencé-Girault S, Lebreton S, Bunau O, Dang P, Bargain F. Simultaneous SAXS-WAXS experiments on semi-crystalline polymers: Example of PA11 and its brill transition. *Crystals* **9**, 271 (2019).
19. Gong J, McGaughey AJ. Device - level thermodynamic model for an electrocaloric cooler. *International Journal of Energy Research* **44**, 5343-5359 (2020).

REVIEWER COMMENTS

Reviewer #2 (Remarks to the Author):

Authors have adequately addressed most of my concerns, thus I would recommend it for publication without further revision. Thanks.

Reviewer #3 (Remarks to the Author):

The authors have replied on the issues raised by me and the other Reviewers comprehensively. However, some of the issues are still not resolved. Provided that the authors can reply on the following points, the manuscript can be considered again for publication.

1) To the 3rd issue that I raised regarding the interface effect that arises between the low-k filler and the polymer matrix, the authors replied by using a simulation which makes use of Maxwell relations, and they show that electric field distortion also exists at the interface between a low-k filler and the polymer. The authors are advised to include the details of the simulation in the Supplementary Information.

2) To the 2nd issue that I raised regarding the WAXD spectra of the materials, the authors included a new Figure as Supplementary Fig. 13, which shows deconvolution of a peak in polar, non-polar and amorphous peaks. Is this peak the most representative peak to do this deconvolution process? The authors can add a sentence explaining why they have chosen this peak by citing previous literature (preferably not from their own). The same peak is used in Ref.11. However, the peak position is considerably different between ref.11 and this study (Fig. S17 of ref.11 shows that alpha phase is close 15 degrees, while in this manuscript Fig. S13, alpha phase is around 12 degrees). What is the origin of this peak position difference?

3) To the 7th issue that I raised regarding the beta coefficient, the authors responded by adequately explaining the terms. Now, the equation number is changed so they need to correct this on Page 17. How did the authors obtain the equation 2 in the revised manuscript? This is not clear from ref. 59 they cited in the revised manuscript. The authors can include this derivation in the Supplementary Information. And it is still not clear how they obtained the beta coefficient by using fits to the dielectric constant.

4) In their reply to reviewer 1, the authors explain that thermal conductivity decreases back after the increase until 2.6% ND content. They attribute this decrease to the pronounced agglomeration of ND particles. The authors can include this observation in the Supplementary Information. Does the same decrease occur also in electrical properties such as dielectric constant, polarization and ECE?

5) The authors mention that space charge-induced conduction losses may contribute to the polarization and use a method to subtract them from the total polarization to obtain the 'real' polarization. Why did they prefer the method they used over the more commonly used PUND (Positive-Up Negative-Down) method? Also, regarding the space charges, how do the authors make sure that the electric potential build-up that they measure at the interface between the polymer and the ND using EFM does not originate from the space charges that can build-up when two electrically dissimilar materials (in this case the polymer and ND) are brought in touch?

Response to the Reviewers' Comments

Reviewer #2 (Remarks to the Author):

Authors have adequately addressed most of my concerns, thus I would recommend it for publication without further revision. Thanks.

Response: We thank the reviewer so much for acknowledging our work.

Reviewer #3 (Remarks to the Author):

The authors have replied on the issues raised by me and the other Reviewers comprehensively. However, some of the issues are still not resolved. Provided that the authors can reply on the following points, the manuscript can be considered again for publication.

1) To the 3rd issue that I raised regarding the interface effect that arises between the low-k filler and the polymer matrix, the authors replied by using a simulation which makes use of Maxwell relations, and they show that electric field distortion also exists at the interface between a low-k filler and the polymer. The authors are advised to include the details of the simulation in the Supplementary Information.

Response: We thank the reviewer for the comment. In the response to issue #3, we utilized the Maxwell equations, rather than the Maxwell relations. The Maxwell equations (Eq. R1-R4,) define the electromagnetic (EM) fields and their properties inside a media. On the other hand, Maxwell relations are frequently utilized to estimate the ECE for normal ferroelectrics (which are not suitable for the relaxor ferroelectric polymer we studied here). We should do better to highlight the simulation method we used here because the two methods are easily confused by their similar names but the

two are actually for different purposes.

$$\nabla \cdot \mathbf{D} = \rho_V \quad (\text{Eq. R1})$$

$$\nabla \cdot \mathbf{B} = 0 \quad (\text{Eq. R2})$$

$$\nabla \times \mathbf{E} = -\frac{\partial \mathbf{B}}{\partial t} \quad (\text{Eq. R3})$$

$$\nabla \times \mathbf{H} = -\frac{\partial \mathbf{D}}{\partial t} + \mathbf{J} \quad (\text{Eq. R4})$$

where \mathbf{D} is the electric flux density and ρ_V is the electric volume charge density. The magnetic flux density (\mathbf{B}) is related to the magnetic field (\mathbf{H}). In addition, \mathbf{E} represents the electric field and \mathbf{J} is the electric current density.

Our simulations demonstrate the electric field will be distorted at any interface as long as two sides have different permittivity. We have added the simulation details of electric field distortion to **Supplementary Section 2.12**. We used the finite element software COMSOL 5.5 to simulate the distribution of electric field lines in the composite material under an electric field. In the simulation, the diameter of the particle is 50 nm, surrounded by the base terpolymer, and the entire area is a square with a side length of 100 nm. Here the dielectric constant of the base terpolymer is set to 42.

The low-k filler chosen here is ND, which has a dielectric constant near room temperature of about 6. And the high-k filler is BST, with a dielectric constant near room temperature of about 1000. A voltage of 10V was applied to both ends of the nanocomposite (*i.e.*, an electric field strength of $100 \text{ MV} \cdot \text{m}^{-1}$). The electric field distortion around a spherical filler with different permittivity is shown in **Fig. R1**.

Fig. R1. Schematic electric field distortion around a spherical filler with a permittivity higher **(a)** or lower **(b)** than the polymeric matrix. The applied electric field is $100 \text{ MV} \cdot \text{m}^{-1}$.

2) To the 2nd issue that I raised regarding the WAXD spectra of the materials, the authors included a new Figure as Supplementary Fig. 13, which shows deconvolution of a peak in polar, non-polar and amorphous peaks. Is this peak the most representative peak to do this deconvolution process? The authors can add a sentence explaining why they have chosen this peak by citing previous literature (preferably not from their own). The same peak is used in Ref. 11. However, the peak position is considerably different between ref. 11 and this study (Fig. S17 of ref. 11 shows that alpha phase is close 15 degrees, while in this manuscript Fig. S13, alpha phase is around 12 degrees). What is the origin of this peak position difference?

Response: We thank the reviewer for the comment. This way of analyzing peaks has been confirmed in several independent laboratories^{1, 2}. Wide-angle X-ray scattering (WAXS) provided us with insights into the lateral coherence of the crystalline moieties in pristine and ND-incorporated nanocomposites. Here we focused on the Bragg peak resulting from the juxtaposition of the two diffraction lines resulting from the (200) and (110) planes, which can be associated with the inter-planar distance, d_{200}/d_{110} , of the orthorhombic pseudo-hexagonal phase, often referred to as RFE (Relaxor Ferroelectric) structure, as described in previous works^{1, 3}. We've added the description to page 10 and page 11 in the manuscript. (marked in red)

The tests of WAXD carried out in the study and Ref. 11 in the manuscript⁴ **were done in the different line stations** at the Shanghai Synchrotron Radiation Facility (SSRF). The wavelength of the X-ray in this study was 1.03 Å and the wavelength of the X-ray in Ref. 11 was 1.24 Å. According to the Bragg equation, a smaller X-ray wavelength corresponds to a smaller angle between the incident beam and the reflecting surface.

$$2d_s \sin \theta = n\lambda \quad (\text{Eq. R5})$$

where d_s is the crystal plane spacing, θ is the angle between the incident beam and the reflecting plane, λ is the X-ray wavelength and n is the number of diffraction levels. This can explain the alpha phase in our study is around 12°.

3) To the 7th issue that I raised regarding the beta coefficient, the authors responded by adequately explaining the terms. Now, the equation number is changed so they need to correct this on Page 17. How did the authors obtain the equation 2 in the revised manuscript? This is not clear from ref. 59 they cited in the revised manuscript. The authors can include this derivation in the Supplementary Information. And it is still not clear how they obtained the beta coefficient by using fits to the dielectric constant.

Response: Thanks for the reviewer's kind suggestions. We've made a correction to the equation number on page 16 in the main text. In addition, the derivation of **Equation 2** in the manuscript has also been added to the **Supplementary Section 2.14**. (Marked in red on page 24-25)

ECE in an electrical insulating material can be estimated from Landau-Devonshire (L-D) phenomenological theory⁵. In a ferroelectric material, the Gibbs free energy can be written as an expansion utilizing polarization as the order parameter,

$$G = G_0 + \frac{1}{2}\alpha P^2 + \frac{1}{4}\xi P^4 + \frac{1}{6}\zeta P^6 + \dots - EP \quad (\text{Eq. R6})$$

where P is the polarization, $\alpha = \beta(T - T_0)$, and β , ξ and ζ are phenomenological coefficients that are temperature independent.

In polar materials, the Gibbs free energy can be written as

$$G = U - TS - X_i x_i - E_j D_j \quad (\text{Eq. R7})$$

where U is the internal energy, T the temperature, S the entropy, X the stress, x the strain, E the electric field and D the electric displacement. Einstein notation i runs from 1 to 6 and j from 1 to 3.

Written in the differential form, the Eq. R7 is

$$dG = -S dT - x_i dX_i - D_j dE_j \quad (\text{Eq. R8})$$

For most electrocaloric (EC) materials investigated, $P \approx D$. Due to $S = -\left(\frac{\partial G}{\partial T}\right)_{E,X}$, one obtains the isothermal entropy change of the system ΔS and the adiabatic temperature change ΔT_{EC} under an electric field E_H and $E_L=0$,

$$\Delta S = -\frac{1}{2}\beta[P^2(E_H) - P^2(E_L)] = -\frac{1}{2}\beta P^2 \quad (\text{Eq. R9})$$

We obtained the β coefficient by utilizing the experimental data (maximum polarization and the measured entropy changes), aiming to evaluate how efficient the overall polarization in generating ECE. The β coefficient we reported here did not come from the fitting of the temperature-dependent permittivity. Conventional Curie-Weiss law works well with the normal ferroelectrics. For relaxor ferroelectrics, Pric et al.⁶ has shown that the Curie-Weiss law should be modified, the β coefficient cannot be obtained using the same way. If we simply apply the conventional Curie-Weiss law, the β coefficients for the terpolymer and the T-ND-2.6% were similar, around $1.5 \times 10^4 \text{ J} \cdot \text{m}^4 \cdot \text{kg}^{-1} \cdot \text{K}^{-1} \cdot \text{C}^{-2}$, whereas we observed that the β coefficient was slightly reducing with increasingly incorporating the ND nanoparticles by utilizing $\Delta S = -\frac{1}{2}\beta P^2$ (Fig. 3d in the main text). We believe that there is still rich physics embedded in the relaxor ferroelectric polymers and their ECE, and we are looking forward to working with physicists in this field in the near future.

4) In their reply to reviewer 1, the authors explain that thermal conductivity decreases back after the increase until 2.6% ND content. They attribute this decrease to the pronounced agglomeration of ND particles. The authors can include this observation in the Supplementary Information. Does the same decrease occur also in electrical properties such as dielectric constant, polarization and ECE?

Response: Thanks for the reviewer's suggestions. Our response to reviewer 1 was about the relationship between cooling performance and ND content (shown in **Fig. R7** in the first version (**Fig. R3** in the current version) of the response to the comments). We observed an increasing thermal conductivity of the ND-incorporated nanocomposites with an increasing the content of the ND (see in **Fig. R2**).

Fig. R2. The thermal conductivity of the base terpolymer and ND-incorporated nanocomposites.

Different from trend of the thermal conductivity, further increasing in NDs fillers (T-ND-5.4%) will lead to a sharp decrease in the ECE (see **Fig. R3**) of the ND-incorporated nanocomposites due to more pronounced agglomeration (**Fig. R4**).

Fig. R3. The ECE of the base terpolymer and ND-incorporated nanocomposites. **(a)** EC-induced ΔS . **(b)** EC-induced ΔT .

Fig. R4. SEM of NDs. **(a)** agglomerated distribution in T-ND-5.4%. **(b)** Even distribution in T-ND-2.6%.

As precisely pointed out by the reviewer, further increase in NDs fillings will lead to a sharp decrease in the polarization (essentially the permittivity at high field) of the ND-incorporated nanocomposites (**Fig. R5**), which exhibited a similar trend to the ECE. We have added the overall performance at high particle content to the **Supplementary Section 2.15**. (Marked in red on page 25-27)

Fig. R5. The high-field polarization of the base terpolymer and ND-incorporated nanocomposites.

5) *The authors mention that space charge-induced conduction losses may contribute to the polarization and use a method to subtract them from the total polarization to obtain the 'real' polarization. Why did they prefer the method they used over the more commonly used PUND (Positive-Up Negative-Down) method? Also, regarding the space charges, how do the authors make sure that the electric potential build-up that they measure at the interface between the polymer and the ND using EFM does not originate from the space charges that can build-up when two electrically dissimilar materials (in this case the polymer and ND) are brought in touch?*

Response: We thank the reviewer for the in-depth comments. What we intended to say here is that we removed the conduction loss from the measured P-E loop, but we referred to it as the space-charge-induced polarization by mistake in the original draft. This has been corrected in the first revision. What we wanted to subtract, was the conduction loss which would affect the maximum polarization.

For the P(VDF-TrFE-CFE), which is considered a typical relaxor ferroelectric

material, the remnant polarization should be zero, as there should be no spontaneous polarization. For the polarization measurement at low frequencies such as 1~10 Hz, the conduction loss will strongly participate in the measured results. Therefore, we observed the polarization value at the electric field $E=0$, which should be subtracted to obtain the conduction-free polarization, as shown in **Supplementary Section 2.8**. The method has been widely used for relaxor ferroelectrics^{7, 8}, and the results should be equivalent to the PUND test at high frequency. At the low frequencies (1~10 Hz), however, the PUND method would not be able to subtract the conduction loss but was regularly applied to extract the remnant polarization, which is none for the relaxor. Therefore, the PUND should show a similar result to the regular PE, for the relaxor ferroelectric polymer we studied here (**Fig. R6**). We noted that the method is under the assumption that the conduction loss is linearly occurring with the ramping up and down of the electric field. For the field of $100 \text{ MV}\cdot\text{m}^{-1}$, which is not quite high for the polymers, the assumption is valid⁷.

Based on the reviewer's suggestion, we performed PUND tests on the ND-incorporated nanocomposites, and the results are shown in **Fig. R6**. PUND tests were measured using a modified Sawyer-Tower circuit (The pulse width is 100 ms and the pulse delay is 1 ms).

Fig. R6. PUND measurement of ND-incorporated nanocomposites, polarization vs time.

Switchable polarization can be calculated as,

$$dP = P^* - P^ \quad (\text{Eq. R10})$$

where P^* = switchable + non-switchable polarization. P^\wedge = non-switchable polarization and Q_{switched} is switched charged density⁹. Substituting these values of P^* and P^\wedge in Eq. R8, true switched charged density has been calculated to be $0.009 \text{ C}\cdot\text{m}^{-2}$, which is quite small when compared with the peak polarization, which is understandable considering the relaxor P(VDF-TrFE-CFE) should not exhibit hysteresis polarization (P_r). As a result, the polarization peaks obtained from the PUND test are almost identical to those obtained by our correction method (**$0.065 \text{ C}\cdot\text{m}^{-2}$ vs $0.066 \text{ C}\cdot\text{m}^{-2}$ at an amplitude of $100 \text{ MV}\cdot\text{m}^{-1}$, see Fig. R7**). On the other hand, the resulting PE (PUND) cannot subtract all the conduction loss considering the relatively long duration of the application of the field (**Fig. R7**). Therefore, to subtract the conduction loss, we used the method described in **Supplementary Section 2.8**, which are frequently used for separating the effect of conduction loss from the maximum polarization^{7,8}.

Fig. R7. PUND measurement of ND-incorporated nanocomposites, polarization vs electric field.

The reviewer is correct that surface charge accumulation could result in artifacts in the EFM imaging. In the test, we did two procedures that address the issue. 1. The probe was scanning at a lifted height and the tip does not touch the sample when applying voltage to the tip, therefore the induced electrostatic charge accumulation on the surface of the sample is limited. 2. We used an ionizing air blower (Tronovo, TR7001) to blow

on our samples for half an hour to eliminate the effects of surface static electricity before conducting the EFM test. These procedures are frequently used in the previously reported studies^{10, 11}. Ref 10 explains in Supplementary Note 5 why they made the assumption of electrostatic charge accumulation on the surface of sample (Procedure 1). Ref 11 (Section 2) mentioned that the specimen was cleaned with anhydrous alcohol and dried by an ionizing air blower in order to excluding the effect of accumulated charge (Procedure 2). Therefore, we believe that the test obtained is the electric potential build-up due to the interfacial effect of the nanocomposites, rather than being influenced by the space charge.

1. Le Goupil F, *et al.* Enhanced Electrocaloric Response of Vinylidene Fluoride–Based Polymers via One-Step Molecular Engineering. *Advanced Functional Materials* **31**, 2007043 (2021).
2. Chen X, *et al.* Relaxor ferroelectric polymer exhibits ultrahigh electromechanical coupling at low electric field. *Science* **375**, 1418-1422 (2022).
3. Bargain F, Thuau D, Panine P, Hadziioannou G, Dos Santos FD, Tence-Girault S. Thermal behavior of poly (VDF-ter-TrFE-ter-CTFE) copolymers: Influence of CTFE termonomer on the crystal-crystal transitions. *Polymer* **161**, 64-77 (2019).
4. Qian X, *et al.* High-entropy polymer produces a giant electrocaloric effect at low fields. *Nature* **600**, (2021).
5. Lines ME, Glass AM. *Principles and applications of ferroelectrics and related materials*. Oxford university press (2001).
6. Pirc R, Kutnjak Z, Blinc R, Zhang Q. Electrocaloric effect in relaxor ferroelectrics. *Journal of Applied Physics* **110**, (2011).
7. Chu B. *PVDF-based copolymers, terpolymers and their multi-component material systems for capacitor applications*. The Pennsylvania State University (2008).
8. Chu B, Zhou Y, Zhang S. Charging and Discharging Characteristics of Dielectric Polymer Materials. In: *Dielectric Polymer Materials for High-Density Energy Storage*. Elsevier (2018).

9. Pokhriyal P, *et al.* Possibility of relaxor-type ferroelectricity in delafossite CuCrO₂ near room temperature. *Solid State Sciences* **112**, 106509 (2021).
10. Simin, *et al.* Direct Detection of Local Electric Polarization in the Interfacial Region in Ferroelectric Polymer Nanocomposites. *Advanced Materials* **31**, 1807722-1807722 (2018).
11. Zhang J-w, Cao D-k, Yin X-q, Putson C, Xiao P, Wang Q. Investigation on the effect of accumulated charge-induced degradation on multilayer photovoltaic insulating backsheets based on atomic force microscopy. *ACS Applied Energy Materials* **3**, 8946-8952 (2020).

REVIEWER COMMENTS

Reviewer #1 (Remarks to the Author):

The authors have addressed many of my comments and improved the overall clarity and quality of the manuscript. However, I still have a few concerns that need to be addressed.

1. It was obvious that the introduction of ND led to an increase in leakage (Supplementary Fig. 16 and Supplementary Fig. 17.a-c). As described by the authors that “the results indicate that the T-ND-2.6% exhibited no obvious joule heating during the whole time of application of the electric field (100 MV·m⁻¹)” (In line 146 of the Supplementary). The authors should provide data related to the variation of leakage current density with electric field and discuss Joule heating.

2. From the author's perspective, the above analysis suggested that the origin of the consequently enhanced ECE was different from the polar high entropy polymers. The enhanced ECE might be attributed to the increased polarization that was induced at the internal interfaces. However, further increasing in NDs fillers (T-ND-5.4%) would lead to a sharp decrease in the ECE (see Supplementary Fig. 25) of the ND-incorporated nanocomposites. The authors concluded that the attenuation of the ECE was caused by nanoparticle agglomeration. As shown in Supplementary Fig. 21 (TEM images of T-ND-2.6%), agglomerates were always present in this system. Therefore, this should not be the main reason for the decline of the ECE in the T-ND-5.4%. If the enhanced ECE was caused by an increase in polarization induced by the internal interface, the ECE should be enhanced rather than decreased in the T-ND-5.4% due to the increased internal interface. So what was the real origin of the enhanced ECE by NDs fillers?

Reviewer #3 (Remarks to the Author):

The authors responded to all the remaining points I raised after the 1st revision and also attempted some new measurements. While some minor issues still remain (as I explain in the last paragraph), the manuscript can now be accepted for publication. However, I think that the language of the manuscript should be improved.

E.g. In the newly added sentence on Page 14, line 260, ‘Therefore, the β coefficient is essentially describing how efficient the maximum polarization is in generating the ECE. ‘is’ was missing.

Or in the abstract, the sentence starting on line 22 is grammatically problematic. The authors are suggested to polish the language as much as possible throughout the manuscript.

Page 16, line 302: '...field-induced...'

Another final remark: In their response letter, the authors replied to the 5th point that the conduction loss was referred to as space-charge-induced polarization by mistake. However, in the revised manuscript on page 13, line 283, it is still written 'we observed that the space charge-induced losses.'

Regarding the space charges: Actually, the authors seem to have misunderstood my question regarding the space charges that might have formed at the polymer-dielectric interface. I was not asking about the possible surface charge accumulation. What I was asking about was the origin of the increased potential at the interface observed by EFM. I asked whether that potential increase might originate from the space charges that might form at the interface. The authors concluded their reply by stating that '...the electric potential build-up due to the interfacial effect of the nanocomposites, rather than being influenced by space charge.' Then, the question remains: what exactly do they understand from the 'interfacial effect' of the nanocomposites? I know that there are many different explanations about this in the literature and no consensus is reached and therefore I don't find it necessary to ask further questions at this point.

Response to the Reviewers' Comments

Reviewer #1 (Remarks to the Author):

The authors have addressed many of my comments and improved the overall clarity and quality of the manuscript. However, I still have a few concerns that need to be addressed.

1. It was obvious that the introduction of ND led to an increase in leakage (Supplementary Fig. 16 and Supplementary Fig. 17.a-c). As described by the authors that “the results indicate that the T-ND-2.6% exhibited no obvious joule heating during the whole time of application of the electric field (100 MV·m⁻¹)”(In line 146 of the Supplementary). The authors should provide data related to the variation of leakage current density with electric field and discuss Joule heating.

Response: Thanks for the reviewer's suggestions. We measured the leakage current density of our films to evaluate their Joule heating during the ECE measurement¹. The variation of leakage current density with electric field is shown in **Fig. R1**.

Fig. R1. Variation of leakage current density with electric field of the base terpolymer and T-ND-2.6%.

T-ND-2.6% shows a slight increase (< 30%) of leakage current density compared to

the base terpolymer. We then use the leakage data to calculate the equivalent Joule heating when conducting ECE measurement (see **Fig. R2a**). At 100 MV m^{-1} , T-ND-2.6% produces a Joule heating of 171.27 kJ m^{-3} , sharing less than 1% of the observed EC effect ($28582.2 \text{ kJ m}^{-3}$).

In addition, we directly tested the heat flux across the EC films with a heat flux sensor (see **Fig. R2b**). The results indicate that the T-ND-2.6% exhibited negligible Joule heating during the whole time of application of the electric field (100 MV m^{-1}). Both the leakage current measurement and the heat flux signal of EC cycles confirm that the enhanced conduction loss by adding small amount of ND is limited and would not affect the EC performance of the materials. We have added the discussions to **Supplementary Section 2.4**.

Fig. R2. Joule heating (**a**) and Heat flow intensity (**b**) of the base terpolymer and T-ND-2.6%.

2. From the author's perspective, the above analysis suggested that the origin of the consequently enhanced ECE was different from the polar high entropy polymers. The enhanced ECE might be attributed to the increased polarization that was induced at the internal interfaces. However, further increasing in NDs fillers (T-ND-5.4%) would lead to a sharp decrease in the ECE (see Supplementary Fig. 25) of the ND-incorporated nanocomposites. The authors concluded that the attenuation of the ECE was caused by nanoparticle agglomeration. As shown in Supplementary Fig. 21 (TEM images of T-ND-2.6%), agglomerates were always present in this system. Therefore,

this should not be the main reason for the decline of the ECE in the T-ND-5.4%. If the enhanced ECE was caused by an increase in polarization induced by the internal interface, the ECE should be enhanced rather than decreased in the T-ND-5.4% due to the increased internal interface. So, what was the real origin of the enhanced ECE by NDs fillers?

Response: We thank the reviewer for the valuable comment. In our work, we believe that the effective interfacial region induced by ND particles is the key to enhance the effective polarization and thus improve the ECE of the nanocomposites. However, this does not necessarily mean that the higher filler content will render better EC performances. The reasons can be summarized as:

- (1) The reduction of effective interfacial layer at high filler contents.** It is intuitively thought that higher filler content would provide more interfacial regions, thus facilitating the interfacial effect. However, recent studies have demonstrated that the dielectric constant (or polarization) were increased in a diffused layer surrounding the particles with thickness ranging from tens to hundreds of nanometers, in which the dielectric constant normally distributed^{2, 3, 4}. Excessive fillers would lead to the overlap of the maximized dielectric constant region between every two particles, resulting in a decay of interfacial effect then less enhanced polarization^{2, 3}. This decay is even more significant when particles undergo severely agglomeration (vanishment of the diffused layer). In our work, we observed a similar phenomenon (See **Fig. R3**), where the effective polarization reaches the maximum in T-ND-2.6% then decays at T-ND-5.4%. The SEM images also suggest the particles in T-ND-5.4% are more likely to aggregate into large clumps (See **Fig. R4**). The results match well with the newly developed dielectric constant model in polymeric composites^{2, 3}. In other words, the decrease of effective polarization at T-ND-5.4% leads to the decrease of ECE.

Fig. R3. ECE (a) and Polarization (b) of the base terpolymer and ND-incorporated nanocomposites.

Fig. R4. SEM of ND-incorporated nanocomposites. (a) agglomerated distribution in T-ND-5.4%. (b) Relatively even distribution in T-ND-2.6%.

(2) **The greatly enhanced conduction loss at high filler contents.** The conduction loss of a dielectric polymeric nanocomposites can be sharply enhanced with the increase of filler content, leading to a large sacrifice in EC cooling capacity. As shown in **Fig. R5**, compared to the base terpolymer and T-ND-2.6%, T-ND-5.4% exhibits a significant Joule heating caused by the conduction, which can lead to a shift of baseline (gap between two equilibrium states). Once the electric fields are removed, the EC cooling will first compensate the residual Joule heating flux and then form a negative peak with a reduced intensity. Therefore, although adding more fillers might provide more particle surface, the highly increased conduction loss would sharply reduce the effective cooling performance of the EC nanocomposites.

Fig. R5. Heat flow intensity of the base terpolymer and ND-incorporated nanocomposites.

In conclusion, we still confirm the interfacial-induced enhanced polarization accounts for the enhanced ECE. Thanks to the reviewer's suggestion, and we have added the discussions to **Supplementary Section 2.15**.

Reviewer #3 (Remarks to the Author):

The authors responded to all the remaining points I raised after the 1st revision and also attempted some new measurements. While some minor issues still remain (as I explain in the last paragraph), the manuscript can now be accepted for publication. However, I think that the language of the manuscript should be improved.

E.g. In the newly added sentence on Page 14, line 260, 'Therefore, the β coefficient is essentially describing how efficient the maximum polarization is in generating the ECE. 'is' was missing.

Or in the abstract, the sentence starting on line 22 is grammatically problematic. The authors are suggested to polish the language as much as possible throughout the manuscript.

Page 16, line 302: '...field-induced...'

Another final remark: In their response letter, the authors replied to the 5th point that the conduction loss was referred to as space-charge-induced polarization by mistake. However, in the revised manuscript on page 13, line 283, it is still written 'we observed that the space charge-induced losses.'

Response: We sincerely thank the reviewer for acknowledging our work. Based on the suggestions, we have carefully gone through the manuscript and corrected all the typos and errors. (marked in red on abstract, page 13, 14 and 16)

Regarding the space charges: Actually, the authors seem to have misunderstood my question regarding the space charges that might have formed at the polymer-dielectric interface. I was not asking about the possible surface charge accumulation. What I was asking about was the origin of the increased potential at the interface observed by EFM. I asked whether that potential increase might originate from the space charges that might form at the interface. The authors concluded their reply by stating that '...the electric potential build-up due to the interfacial effect of the nanocomposites, rather than being influenced by space charge.' Then, the question remains: what exactly do they understand from the 'interfacial effect' of the nanocomposites? I know that there

are many different explanations about this in the literature and no consensus is reached and therefore I don't find it necessary to ask further questions at this point.

Response: We appreciate the reviewer for his/her patience on clearly illustrating the concerns, and initially we did misunderstand the comments.

For EFM, the increased potential in the interfacial region detected by EFM can be caused by multiple possibilities. The major two are (1) large enhancement of local dielectric constant that influences the electrostatic force^{5, 6}; (2) change of surface charges⁷. Indeed, as the reviewer mentioned, the space charges in/on the samples can also affect the surface charge that may change the EFM surface potential. We are unable to fully exclude which one mainly accounts for the interfacial effect in our studies. But as precisely pointed out by the reviewer, there are many explanations about the interfacial effect of the nanocomposites, so the answers could be multiple reasons.

However, we would like to emphasize that no matter which process is dominant, both of them can lead to the enhanced polarization as we observed in the Polarization-Electric field measurements, thus still corroborating our argument that ND-incorporated nanocomposites with large improvement in ECE is caused by the enhanced polarization at the interface regions. Thanks to the reviewer's suggestion and we have added the discussions to **Supplementary Section 2.10**.

There still are rich physics embedded in the “interfacial effect”, and we are looking forward to working with physicists in this field and leveraging additional characterization tools in the near future.

Reference

1. Lu S, *et al.* Joule heating-A significant factor in electrocaloric effect. *Ceramics International* **45**, 16992-16998 (2019).
2. Li L, *et al.* Significant improvements in dielectric constant and energy density of ferroelectric polymer nanocomposites enabled by ultralow contents of nanofillers. *Advanced Materials* **33**, 2102392 (2021).

3. Thakur Y, *et al.* Enhancement of the dielectric response in polymer nanocomposites with low dielectric constant fillers. *Nanoscale* **9**, 10992-10997 (2017).
4. Zhang B, Chen X, Lu W, Zhang Q, Bernholc J. Morphology-induced dielectric enhancement in polymer nanocomposites. *Nanoscale* **13**, 10933-10942 (2021).
5. Peng S, Zeng Q, Yang X, Hu J, Qiu X, He J. Local dielectric property detection of the interface between nanoparticle and polymer in nanocomposite dielectrics. *Scientific reports* **6**, 38978 (2016).
6. Simin, *et al.* Direct Detection of Local Electric Polarization in the Interfacial Region in Ferroelectric Polymer Nanocomposites. *Advanced Materials* **31**, 1807722-1807722 (2018).
7. Schoenherr P, *et al.* Observation of uncompensated bound charges at improper ferroelectric domain walls. *Nano letters* **19**, 1659-1664 (2019).

REVIEWERS' COMMENTS

Reviewer #1 (Remarks to the Author):

For the current version, it could be published with the following corrections.

On the basis of the existing research methodology, the authors relied only on single-line experimental evidence of polarization enhancement to demonstrate that the ND-enhanced electrocaloric effect came from interface polarization. In order to ensure the accuracy of the communication of scientific knowledge, it was recommended that the author reduced the affirmative discussion of this part.e.g.

Page 6, lines 112-115, to verify the contribution

Page 24, lines 448-450, we observed that the low-k ND could also.....ferroelectric counterparts.

Response to the Reviewers' Comments

Reviewer #1 (Remarks to the Author):

For the current version, it could be published with the following corrections.

On the basis of the existing research methodology, the authors relied only on single-line experimental evidence of polarization enhancement to demonstrate that the ND-enhanced electrocaloric effect came from interface polarization. In order to ensure the accuracy of the communication of scientific knowledge, it was recommended that the author reduced the affirmative discussion of this part.e.g.

Page 6, lines 112-115, to verify the contribution

Page 24, lines 448-450, we observed that the low-k ND could also.....ferroelectric counterparts.

Response: Thanks for the reviewer's suggestions. We have modified the discussion to ensure the accuracy of our expression. (Marked red in Page 6 and Page 24 in the manuscript.)